# Specific facial signals associate with categories of social actions conveyed through questions

**Naomi Nota**[1,2]*, **James P. Trujillo**[1,2], **Judith Holler**[1,2]

**1** Donders Institute for Brain, Cognition, and Behaviour, Radboud University, Nijmegen, The Netherlands,
**2** Max Planck Institute for Psycholinguistics, Nijmegen, The Netherlands

\* Naomi.Nota@donders.ru.nl

**Data Availability Statement:** Raw data cannot be shared publicly because it consists of video recordings of participants. The preregistration for this study is available on the As Predicted website

## Abstract

The early recognition of fundamental social actions, like questions, is crucial for understanding the speaker's intended message and planning a timely response in conversation. Questions themselves may express more than one social action category (e.g., an information request "What time is it?", an invitation "Will you come to my party?" or a criticism "Are you crazy?"). Although human language use occurs predominantly in a multimodal context, prior research on social actions has mainly focused on the verbal modality. This study breaks new ground by investigating how conversational facial signals may map onto the expression of different types of social actions conveyed through questions. The distribution, timing, and temporal organization of facial signals across social actions was analysed in a rich corpus of naturalistic, dyadic face-to-face Dutch conversations. These social actions were: Information Requests, Understanding Checks, Self-Directed questions, Stance or Sentiment questions, Other-Initiated Repairs, Active Participation questions, questions for Structuring, Initiating or Maintaining Conversation, and Plans and Actions questions. This is the first study to reveal differences in distribution and timing of facial signals across different types of social actions. The findings raise the possibility that facial signals may facilitate social action recognition during language processing in multimodal face-to-face interaction.

## Introduction

Recognizing social actions is a crucial aspect of having a successful conversation, since they indicate what the utterance 'does' (e.g., performing a request; comparable to 'speech acts' [1, 2]). Early recognition of the social action of an utterance allows next speakers to plan their turn in advance [3–7], thus enabling the fast exchanges of speaking turns seen in typical conversation [8, 9]. In conversation, successfully identifying a turn's social action enables the next speaker to provide an appropriate response. For example, an appropriate response to a question indicating troubles of understanding ("She did what?" [10]) is repair ("I said she did not vote."). Misunderstanding the social action could lead to wrongly interpreting the request for repair as a stance or sentiment question used to express disapproval or criticism (i.e., equivalent to saying "The fact she did not vote is wrong."). It is therefore important for the listener to decipher which kind of social action the question is performing in order to provide a pragmatically appropriate response, and to do so quickly.

https://aspredicted.org/6VZ_L2K. A comprehensive preregistration, anonymized data, the analysis script with additional session information, and supplementary materials can be found on the Open Science Framework project website https://osf.io/u59kb/?view_only=d2b7f98f7ba646d69c8afd5cf09e4b2e.

**Funding:** This work was supported by an ERC Consolidator grant https://erc.europa.eu (#773079, awarded to JH). The funders had no role in study design, data collection and analysis, decision to publish, or preparation of the manuscript.

**Competing interests:** The authors have declared that no competing interests exist.

Research investigating social actions while considering the sequential conversational context has mainly focused on the verbal modality [11–13]. However, human language use occurs predominantly in a multimodal context, including speech and visual bodily signals [6, 14–19]. Speakers often use facial signals during social interaction, and a number of studies showed that (non-emotional) facial signals play a role in marking social actions like questions. Questions are extremely frequent in conversation and fulfil a wide range of fundamental social actions themselves, such as information requests, invitations, offers, criticisms, and so forth [7].

Some studies looked at facial signals with questions performing different social actions, such as information requests [20–26], and echo questions expressing a stance or sentiment such as incredulity [27–29]. Specifically, questions were frequently linked to eyebrow movements like frowns and raises [20–35] as well as direct gaze [21, 36–38]. Common combinations of facial signals have also been associated with social actions [20, 39–42].

Facial signals may be especially beneficial when they occur prior to or early in the verbal utterance to allow quick recognition of the social action. An early timing of facial signals relative to the verbal utterance was observed in several studies [33, 43–45]. Crucially, a recent study analysing a rich corpus of naturalistic dyadic face-to-face conversations revealed that the majority of facial signals happened early in the verbal utterance [33]. Additionally, there were earlier onsets of facial signals in questions compared to responses, and questions occurred with a higher number of facial signals compared to responses. This suggests that early visual marking through facial signals may be most relevant for questions to help fast social action attribution and a quick understanding of the intended message.

Although facial signals may appear early to enable quick recognition of the conveyed message, diverging from this early signalling approach may be meaningful in itself. In Nota et al. [33], mouth movements like smiles were found to often occur relatively late in the utterance. Smiles may signal an ironic or sarcastic intent [39, 46, 47], and these intentions are typically shown at the end of an verbal utterance for a humoristic effect. Therefore, it could be that smiles at the beginning of the utterance convey a different social action compared to smiles at the end, which signal irony or sarcasm. Additionally, the specific temporal organization of facial signals with regard to one another may vary across different social actions. It may be that the specific order that facial signals occur in communicates different social actions of questions.

In sum, although there is some evidence for individual facial signals and common combinations of facial signals associating with specific social actions in conversation, the current study goes beyond previous work by using a data-driven approach on a large dataset of naturalistic dyadic face-to-face conversations to investigate the possibility of a systematic mapping between a range of facial signals and several social actions. Moreover, we study the timing, and temporal organization, of facial signals to determine whether there is a fixed order of facial signals that characterizes different categories of social actions conveyed through questions, including cases where they appear to form social-action-specific clusters of visual signals. The findings will shed light on the extent to which facial signals form a core element of face-to-face language use.

## Current study

Nota et al. [33] found specific distributions and early timings of facial signals in the broad social action category of questions compared to responses. However, since broader social actions in themselves can perform a wide range of different, more specific social actions (as seen above), a much more fine-grained investigation is needed. Here, we investigate facial signals in different social actions of questions using the same corpus of naturalistic, dyadic,

Dutch face-to-face conversations as Nota et al. [33]. To study different social actions of questions, a subset (*n* = 2082) from the transcribed questions from each speaker were coded for their social action category, resulting in eight discrete social action categories of questions. These were 1) *Information Requests*, or requests for new information of a factual or specific nature, of a non-factual or a non-specific nature, elaboration, or confirmation ("What is this?"), 2) *Understanding Checks*, or requests for confirmation about information that was mentioned in the preceding turn or can be inferred from it, or to make sure the interlocutor is following the exchange ("And you said you wanted to travel next week?"; 'CHECK-question' [48]), 3) *Self-Directed questions*, or questions that are not meant for the other speaker, and may fill pauses to show that the speaker wants to keep the turn ("Now where are my keys?"), 4) *Stance or Sentiment questions*, or questions that express humour, disapproval or criticism, seek agreement, compliment, challenge the other speaker to justify or correct something, warn their interlocutor about a problem, or used to make an emphatic remark ("Do you think that is fair?"), 5) *Other-Initiated Repairs*, or questions that seek to resolve mishearings or misunderstandings ("What?", "Who?"), 6) *Active Participation questions*, or news acknowledgments which may or may not encourage elaboration, expressions of surprise, disbelief, or scepticism to what is said by the other speaker, or backchannels ("Oh really?"), 7) questions intended for *Structuring, Initiating or Maintaining Conversation*, or questions checking a precondition for a future action, topic initiations, elaborations, or setting up scenarios ("Guess what?"), and finally 8) *Plans and actions questions*, or proposals for future actions, invitations, suggestions, or offers ("How about lunch together?").

Our main research questions were:

1. What is the distribution of facial signals across social actions?

2. What are the timings of the facial signals with regard to the verbal utterances performing the social actions?

3. What is the temporal organization of facial signals with regard to one another across the different social actions, and are there social action-specific clusters of facial signals?

We hypothesised that social actions would differ with respect to the facial signals they are associated with, since facial signals were previously found to signal social actions [33]. Based on previous literature, we expected an association between *Information Requests* and eyebrow movements such as eyebrow frowns or raises [20–22, 24]. Furthermore, we expected an association between *Self-Directed questions* and gaze shifts, in line with the idea that speakers avert their gaze to signal that they are still in the process of something and do not require active participation of the addressee [39, 42]. Moreover, we expected an association between *Stance or Sentiment questions* and mouth movements, since smiles are used to convey irony [47], and pressed lips are used to express negation or disagreement [40]. We expected an association between *Other-Initiated Repairs* and eyebrow movements such as eyebrow frowns or raises [10, 32, 49]. Backchannels may often be used to convey participation ("No way!?"), therefore, we expected an association between *Active Participation questions* and visual backchannels like eyebrow raises, smiles, pressed lips, and mouth corners down [22, 50]. Echo questions may be used for news acknowledgments, expressions of surprise, or disbelief (e.g., Speaker A: "I'm expecting a baby." Speaker B: "A baby?"), thus, we expected an association between *Active Participation questions* and facial signals used in echo questions like eyebrow raises [29, 30].

In line with Nota et al. [33], and with the idea of early signalling facilitating early action recognition in conversational interaction [3–7], we further hypothesised that most facial signals would occur around the start of the utterance (i.e., eyebrow movements such as frowns, raises, frown raises, eye widenings, squints, blinks, gaze shifts, nose wrinkles). Additionally, we

expected that some facial signals would occur predominantly late in the utterance (i.e., mouth movements such as pressed lips, mouth corners down, and smiles), in agreement with Nota et al. [33].

Lastly, we expected that known combinations of facial signals such as the not-face [40], facial shrug [20, 39, 41], and thinking-face [39, 42] would often co-occur. Due to this study being the first systematic, large-scale analyses of facial signals and social actions, we did not make further social action-specific predictions but instead opted for the data to inform us about the associations.

This study provides new insights into whether facial signals are associated with different social actions performed by questions. This study is primarily exploratory; however, it will lay the groundwork for future experimental investigations in this research area, and allow for more targeted analyses on the contribution of facial signals on social action recognition during language comprehension.

## Methods

A detailed description of the corpus collection, as well as the methods used for social actions transcriptions, facial signals annotations, and interrater reliabilities, can be found in Nota et al. [33] and Trujillo and Holler [51]. The preregistration for this study is available on the As Predicted website https://aspredicted.org/6VZ_L2K. A comprehensive preregistration, depersonalized data, the analysis script with additional session information, and supplementary materials can be found on the Open Science Framework project website https://osf.io/u59kb/.

### Corpus

We based our analyses on recordings of 34 dyads from a corpus of multimodal Dutch face-to-face conversations (CoAct corpus, ERC project #773079 led by JH). These consisted of Dutch native speaker pairs of acquaintances (mean age: 23 ± 8 years, 51 females, 17 males), without motoric or language problems and with normal or corrected-to-normal vision, holding a dyadic casual conversation for one hour while being recorded. There were three parts to the recording session. In the first part, participants held a free conversation. During the second part, participants could discuss statements relating to three different themes: data privacy, social media, and language in teaching. In the third part, participants were instructed to come up with their ideal joint holiday plan. These different sessions were used to elicit a wider range of social actions than may result during the one-hour session when just engaging in the unprompted conversations (e.g., debating pros and cons increasing the chance of agreements and disagreements). We are confident that the data collected in this corpus reflects common everyday dialogue as participants varied in the content they conveyed and expressed diverse perspectives on comparable themes while being recorded, indicating a high level of comfort.

Participants were seated facing each other at approximately 90 cm distance measured from the front edge of the seats. The conversations were recorded in a soundproof room at the Max Planck Institute for Psycholinguistics in Nijmegen, The Netherlands. Two video cameras (Canon XE405) were used to record frontal views of each participant (see Fig 1) at 25 fps.

More video cameras were used to record the scene from different angles, however, for the purpose the current study only the face close-ups were used for best visibility of detailed facial signals. Audio was recorded using two directional microphones (Sennheiser me-64) (see S1 Appendix for an overview of the set-up). The video files and audio files were synchronized and exported from Adobe Premiere Pro CS6 (MPEG, 25 fps) as a single audio-video file per recording session, resulting in a time resolution of approximately 40 ms per frame.

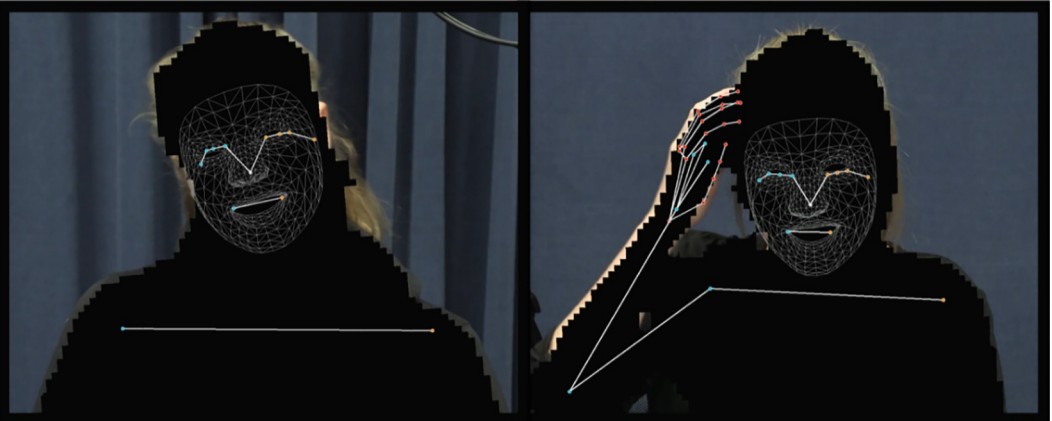

**Fig 1. Still frame from one dyad, showing the frontal camera view used for the present analysis.** Speaker A is shown on the left, Speaker B on the right. Personal identities from the visual recordings were hidden for the purpose of this publication while preserving multimodal information using Masked-Piper [52].

Informed consent was obtained in written form before and after filming. Before the study, participants were asked to fill in a demographics questionnaire. At the end of the study, information was collected about the relationship between the speakers in the dyads and their conversation quality (see S2 Appendix for a summary of the relationship between speakers and their conversation quality, and the complete questionnaire results on the Open Science Framework project website https://osf.io/u59kb/), as well as the Dutch version of the Empathy Quotient [53, 54], the Fear of Negative Evaluation scale [55], and an assessment of explicit awareness of the experimental aim. Information from these questionnaires were collected for a wider range of studies, but are not discussed in the current study, since they were not deemed relevant for the analysis at hand. Participants were compensated with 18 euros. The corpus study was approved by the Ethics Committee of the Social Sciences department of the Radboud University Nijmegen (ethic approval code ECSW 2018–124).

## Transcriptions

Transcriptions of questions, coding of social action categories and facial signals in the corpus were made using ELAN (5.5 [56]).

*Questions.* First, an automatic orthographic transcription of the speech signal was made using the Bavarian Archive for Speech Signals Webservices [57]. All questions were then manually transcribed. The questions were identified and coded largely following the coding scheme of Stivers and Enfield [58]. In order to account for the complexity of the data in the corpus, more rules were applied on an inductive basis, and a holistic approach was adopted that took into consideration visual bodily signals, context, phrasing, intonation, and addressee behaviour. The precise beginnings and endings of the question transcriptions were segmented using Praat (5.1 [59]) based on the criteria of the Eye-tracking in Multimodal Interaction Corpus (EMIC [60, 61]). This resulted in a total of 6778 questions.

Interrater reliability for question identification was calculated with raw agreement [62, 63] and a modified Cohen's kappa using EasyDIAg [64] on 12% of the total data. A standard overlap criterion of 60% was used. This resulted in a raw agreement of 75% and $k = 0.74$ for questions, indicating substantial agreement (for more details, see Nota et al. [33]).

*Social action categories.* A subset of 2082 questions were coded for their social action category. The detailed coding scheme for the social action categories was developed for a larger

**Table 1. Overview of social action categories and their duration for questions in the CoAct corpus included in the present study.**

| Social action | Total number[a] | *Mdn* duration (ms) | *min* duration (ms) | *max* duration (ms) | *IQR* duration (ms) |
|---|---|---|---|---|---|
| InfReq | 695 | 1274 | 241 | 8182 | 1246 |
| UndCheck | 366 | 1324 | 263 | 9476 | 1262 |
| SelfDir | 361 | 1045 | 155 | 6851 | 789 |
| StanSem | 246 | 1320 | 241 | 10143 | 1254 |
| OIR | 126 | 647 | 142 | 3306 | 851 |
| ActPart | 161 | 383 | 169 | 1840 | 240 |
| SIMCo | 74 | 1397 | 426 | 9129 | 1635 |
| PlanAct | 53 | 1571 | 451 | 5918 | 1697 |

*Note. Mdn* = median, *min* = minimum, *max* = maximum, *IQR* = interquartile range, ms = milliseconds.

[a] The total number of social actions differs slightly to Trujillo and Holler [51], due to the coding of four additional social actions at the moment of analysis.

project that the present study is part of, and was inspired by a combination of previous categorizations [10, 12, 58, 65, 66]. We took into account the sequential position and form of the social actions in conversation, state of the common ground between speakers, communicative intention, as well as the result of the speaker's utterance on the addressee. This resulted in eight discrete social action categories: 1) *Information Requests* (InfReq), 2) *Understanding Checks* (UndCheck), 3) *Self-Directed questions* (SelfDir), 4) *Stance or Sentiment questions* (StanSem), 5) *Other-Initiated Repairs* (OIR), 6) *Active Participation questions* (ActPart), 7) Questions intended for *Structuring, Initiating or Maintaining Conversation* (SIMCo), 8) *Plans and Actions questions* (PlanAct). An overview of the social action categories with durations per category is presented in Table 1.

Following the same procedure as for questions transcriptions, interrater reliability for the social action categories was calculated on 686 additionally coded social action categories of the question annotations. This resulted in a raw agreement of 76% and $k = 0.70$, indicating substantial agreement (for more details, see Trujillo and Holler [51]).

*Facial signals.* Facial signals that formed part of the questions coded for social actions were annotated based on the synchronised frontal view videos from the corpus. All of these facial signals involved movements that were judged as carrying some form of communicative meaning related to the questions, as we were interested in the communicative aspect instead of pure muscle movements. Only facial signals that started or ended between a time window of 200 ms before the onset of the question transcriptions and 200 ms after the offset of the question transcriptions were annotated (until their begin or end, which could be outside of the 200 ms time window). These cut off points were agreed based on close qualitative inspection of the data, aiming at a good compromise between accounting for the fact that visual signals can slightly precede or follow the part of speech that they relate to, and trying to avoid including signals which were related to the preceding or following utterance (often spoken by the respective other participant, making them addressee signals) rather than the utterance of interest. Facial signals were coded from where they started until they ended. Movements due to swallowing, inhaling, laughter, or articulation were not considered. No annotations were made when there was insufficient facial signal data due to head movements preventing full visibility or due to occlusions. Lastly, any facial signal annotation that started or ended within 80 ms (two frames) of speech that was unrelated to the question was excluded from the analysis, to reduce the likelihood of including facial signals that were related to any speech from the speaker that did not form part of the target question. This procedure resulted in a total of 4134 facial signal annotations, consisting of: eyebrow movements (frowns, raises, frown raises, unilateral raises), eye

**Table 2. Overview of facial signals in questions and their duration.**

| Signal | Total Number | *Mdn* Duration (ms) | *min* Duration (ms) | *max* Duration (ms) | *IQR* Duration (ms) |
|---|---|---|---|---|---|
| Eyebrow frowns | 253 | 1320 | 80 | 13120 | 1800 |
| Eyebrow raises | 482 | 720 | 40 | 20120 | 1240 |
| Eyebrow frown raises | 41 | 1200 | 200 | 7560 | 2440 |
| Eyebrow unilateral raises | 67 | 480 | 160 | 4120 | 680 |
| Eye widenings | 88 | 820 | 120 | 5240 | 770 |
| Squints | 241 | 1160 | 120 | 10240 | 1560 |
| Blinks | 1592 | 280 | 80 | 1320 | 120 |
| Gaze shifts | 818 | 960 | 40 | 8480 | 1120 |
| Nose wrinkles | 27 | 680 | 200 | 1720 | 760 |
| Pressed lips | 13 | 520 | 280 | 920 | 120 |
| Mouth corners down | 12 | 840 | 280 | 2080 | 940 |
| Smiles | 500 | 2520 | 120 | 16000 | 2760 |

*Note. Mdn* = median

*min* = minimum

*max* = maximum

*IQR* = interquartile range

ms = milliseconds.

widenings, squints, blinks, gaze shifts (gaze away from the addressee, position of the pupil), nose wrinkles, and non-articulatory mouth movements (pressed lips, corners down, smiles). An overview of the facial signals linked to the question transcriptions with durations can be found in Table 2.

A similar procedure as for questions and social action transcriptions was used to calculate interrater reliability on approximately 1% of the total data. In addition, we computed convergent reliability for annotation timing by using a Pearson's correlation (*r*), standard error of measurement (*SeM*), and the mean absolute difference (*Mabs*, in ms) of signal onsets, to assess how precise the annotations were in terms of timing, if there was enough data to compare. All included facial signals from the paired comparisons showed an average raw agreement of 76% and an average kappa of 0.96, indicating almost perfect agreement. When there was enough data to perform a Pearson's correlation, all signals showed $r = 1$ with a $p < .0001$, indicating a strong correlation. There was not enough data to perform a correlation for eyebrow frown raises, nose wrinkles, and mouth corners down. Results are shown in Table 3 (for more details on the facial signals reliability calculations, see Nota et al. [33]).

## Analysis

**Distribution of facial signals across social actions.** The first analyses aimed to quantify and describe the distribution of facial signals across the eight social action categories.

*Associations between facial signals and social action categories.* To study whether social actions predict facial signal distribution, we used generalized linear mixed-effect models (GLMMs). In contrast to the preregistration of this study, in which we intended to include separate GLMMs for each of the 12 facial signals, we performed all comparisons in our main model, since there were not enough data points to perform these models separately for each facial signal. For the facial signals that did have enough data points, results of the separate models can be found in the supplementary materials (https://osf.io/u59kb/). In the current analysis of signal distribution across social actions, we did not differentiate between the

**Table 3. Overview of facial signal reliability scores [33].**

| Signal | agr | k | SeM [a] | Mabs (ms) [a] |
|---|---|---|---|---|
| Eyebrow frowns | 98% | 0.90 | 84.97 | 167.58 |
| Eyebrow raises | 97% | 0.97 | 46.07 | 120.44 |
| Eyebrow frown raises | 100% | 0.83 | 97 | 132 |
| Eyebrow unilateral raises | 99% | 0.88 | 13.49 | 46.57 |
| Eye widenings | 99% | 0.83 | 46.30 | 129.16 |
| Squints | 99% | 0.91 | 29.69 | 73 |
| Blinks | 92% | 0.97 | 9.85 | 30.65 |
| Gaze shifts | 98% | 0.99 | 36.89 | 112 |
| Nose wrinkles | 100% | 0.81 | 24 | 40 |
| Pressed lips | 99% | 0.86 | 34 | 380 |
| Mouth corners down | 97% | 0.80 | 31 | 110 |
| Smiles | 97% | 0.96 | 201.41 | 480.67 |

*Note. agr* = raw agreement [62, 63], *k* = Cohen's kappa [64], *SeM* = standard error of measurement, *Mabs* = mean absolute difference (ms).

[a] One video frame was equivalent to 40 ms. Thus, we considered the variance based on the reliability of the signals shown by *SeM* and *Mabs* as very low (and therefore very precise) when < 40, low when < 80, moderate < 160, and high < 160 (least precise).

different facial signals. Furthermore, the main set of contrasts were corrected for in the main model, therefore, we did not need to apply a Bonferonni correction to adjust the alpha ($\alpha$) threshold.

First, following the recommendations of Meteyard and Davies [67], we fitted the fixed and random parameters of our model on the basis of our research questions. This resulted in the dependent variable facial signal count, with social action and the utterance count per social action as fixed effects. We did not include utterance length as fixed effect, since this analysis was about the overall association between facial signal counts and social actions. We included random intercepts for both signal and item. We did not add random slopes, nor an interaction between potentially modulating factors, because this resulted in overfitting the model. A Poisson distribution was used, which is especially suited for count data that are often highly skewed [68]. To test the significance of the model, we used a likelihood ratio test (ANOVA function) to compare the model of interest to a null model without social action as a predictor, thereby testing whether the variable of interest explained significantly more of the variance than the null model. Furthermore, we performed a post-hoc analysis among social actions after fitting the model, using the Tukey method for comparing eight estimates [69].

*Proportion and rate of facial signals across social actions.* To find out whether a particular facial signal occurred more often in one social action than another, we first calculated how many facial signals of each type occurred together with each social action category out of the respective social action's total number of facial signal occurrences. In contrast to the preregistration, we report the analysis excluding blinks, given that the sheer amount of blinks would overshadow other facial signal distributions, and blinks often serve a clear physiological need to wet the eyes (see supplementary materials for the analysis including blinks). Additionally, we performed an analysis on the rate of facial signals per second. We standardized the amount of facial signal occurrences per social action to utterance length, by dividing by the utterance duration (in sec), to allow for a better comparison between social actions with relatively different utterance lengths. This resulted in facial signal rate collapsed across questions.

## Timings of facial signals within social actions

To determine where facial signal onsets primarily distribute in the verbal utterances, and whether there were differences across social actions, we standardised utterance duration between 0 (onset utterance) and 1 (offset utterance). Facial signal onsets were plotted relative to that number. To enable visualization of facial signal onset distribution before the start of the verbal utterance, the window of analysis was plotted from -1 to 1.

## Temporal organization of facial signals with regard to one another across the different social actions

*Proportion of facial signal sequences.* To capture the sequential patterns of facial signals, we first determined which facial signal sequences were most frequent. We considered a facial signal sequence to consist of at least two (or more) facial signals that occurred in the same verbal utterance. When facial signals occurred simultaneously in the verbal utterance ($< 1\%$), this was transformed to a sequence, and placed in an alphabetical order that depended on the respective facial signal label (e.g., if gaze shifts and eye widenings began at the same time, or eye widenings began before gaze shifts, the sequence would be 'Eye widenings-Gaze shifts' in both cases. However, if gaze shifts began before eye widenings, the sequence would be 'Gaze shifts-Eye widenings'). This means that in certain cases, co-occurring facial signals and facial signals that occur in a sequence could not be distinguished from each other. For convenience, we refer to these instances as sequences. The most frequent facial signal sequences were defined as occurring more than four times in total. Contrary to the preregistration, we did not include plots of frequent sequences and their proportion out of all facial signal sequences. Instead, we wanted to focus on how the frequent sequences distributed across the social actions, but did include the original analysis in the supplementary materials.

To find out how these frequent facial signal sequences distributed across the different social actions, and to see whether there were differences across social actions, we calculated the proportion of the frequent facial signal sequences per social action, out of all sequences in that social action. In contrast to the preregistration, we report the analysis excluding blinks, for the same reason we excluded blinks in prior analyses (see supplementary materials for the analysis including blinks).

*Social action-specific clusters of facial signals.* To see whether groupings of (or single) facial signals could predict an utterance to be one of the eight social action categories, we looked at social action-specific clusters of facial signals by performing a statistical analysis consisting of Decision Tree (DT) models [70]. DT models consist of machine-learning methods to construct prediction models using continuous or categorical data. Based on the input data, DT models build logical "if. . . then" rules to predict the input cases. The models come from partitioning the data space in a recursive way, fitting a prediction model for each partition, which is represented in a DT. In this analysis, partitioning meant finding the specific configuration of facial signal combinations that best predicted whether the utterance was one of the eight social action categories. We used conditional inference (CI [71]) with holdout cross-validation, since CI selects on the basis of permutation significance tests which avoids the potential variable selection bias in similar decision trees and led to the most optimal pruned decision tree. Cross-validation is a technique used to split the data into training and testing datasets, and holdout is the simplest kind as it performs the split only once [72]. We used the default value for distributing the training and validation datasets, resulting in a data fraction of 0.8 for the training dataset, and 0.2 for the validation dataset. In contrast to the preregistration, we report the analysis excluding blinks, for the same reason we excluded blinks in prior analyses. Including blinks led to largely the same results (see supplementary materials for the analysis including

blinks). To test the statistical significance of the classification analysis, we used permutation tests [73]. We used the same data and holdout cross-validation as in the previous classification analysis [33], and repeated the simulation 1000 times.

*Transitional probability between pairs of facial signals over all sequences.* To explore how likely it was that certain facial signals would be adjacent to each other in facial signal sequences (or overlapped) across social actions, we used Markov chains [74, 75]. We first extracted adjacent facial signals from the full set of facial signal sequences (e.g., 'Gaze shifts-Blinks-Eyebrow frowns' became 'Gaze shifts-Blinks' and 'Blinks-Eyebrow frowns'). We then plotted the count of adjacent facial signal pairs from the same utterances over all social actions, as well as per social action category, with the first facial signal of the sequence on the x-axis and next facial signal on the y-axis.

To determine the transitional probability between each pair of facial signals over all sequences, we reshaped the dataframe to a transition matrix and scaled each cell by dividing it by the sum of its row, so that each row was equal to 1. We plotted the transition diagram by excluding transition probabilities below 0.2, in order to make the diagram easier to read. Contrary to the preregistration, we report the transitional probability analysis excluding blinks, since blinks highly skewed our findings (see supplementary materials for the analysis including blinks). Moreover, we did not analyse transitional probability for each social action. This is because not all sequences occurred in each social action, which prevented us from creating a symmetrical matrix for each category. Instead, we analysed transitional probabilities over all social actions, to see whether certain facial signals occur in a specific adjacency pattern in questions more generally.

## Results

### Distribution of facial signals across social actions

Associations between facial signals and social action categories. To determine whether facial signal count was significantly different across social actions, we used GLMMs. We found that social action category significantly predicted facial signal count ($\chi^2(7) = 25.50$, $p < .001$). A post-hoc analysis among social action categories with Tukey-adjusted $p$-values revealed a significantly higher facial signal count in *Information Requests* compared to *Self-Directed questions* (estimate = .15, *SE* = .05, z-ratio = 3.17, $p = .033$), *Other-Initiated Repairs* (estimate = .27, *SE* = .08, z-ratio = 3.27, $p = .024$), and *Active Participation questions* (estimate = .26, *SE* = .08, z-ratio = 3.47, $p = .012$). See Fig 2 for an overview of the model prediction.

### Proportion and rate of facial signals across social actions

To find out whether a facial signal occurred more often with one social action than another, we first looked at the proportion of each facial signal that occurred together with each social action category out of the respective social action's total number of facial signal occurrences. Different distributions of facial signals were found across social actions, such as a high proportion of eyebrow frowns and raises in *Other-Initiated Repairs*, and eyebrow raises in *Active Participation questions* as well as *Plans and Actions questions*. Furthermore, there was a high proportion of gaze shifts away from the addressee in *Self-Directed questions* and questions intended for *Structuring, Initiating or Maintaining Conversation*, and of smiles in *Active Participation questions* and *Stance or Sentiment questions* (Fig 3).

Second, we looked at the rate of facial signals per second across social actions. Different rates of facial signals were found across social actions when taking into account utterance length. For instance, there were high rates of eyebrow raises and eyebrow frown raises in *Other-Initiated Repairs* (Fig 4).

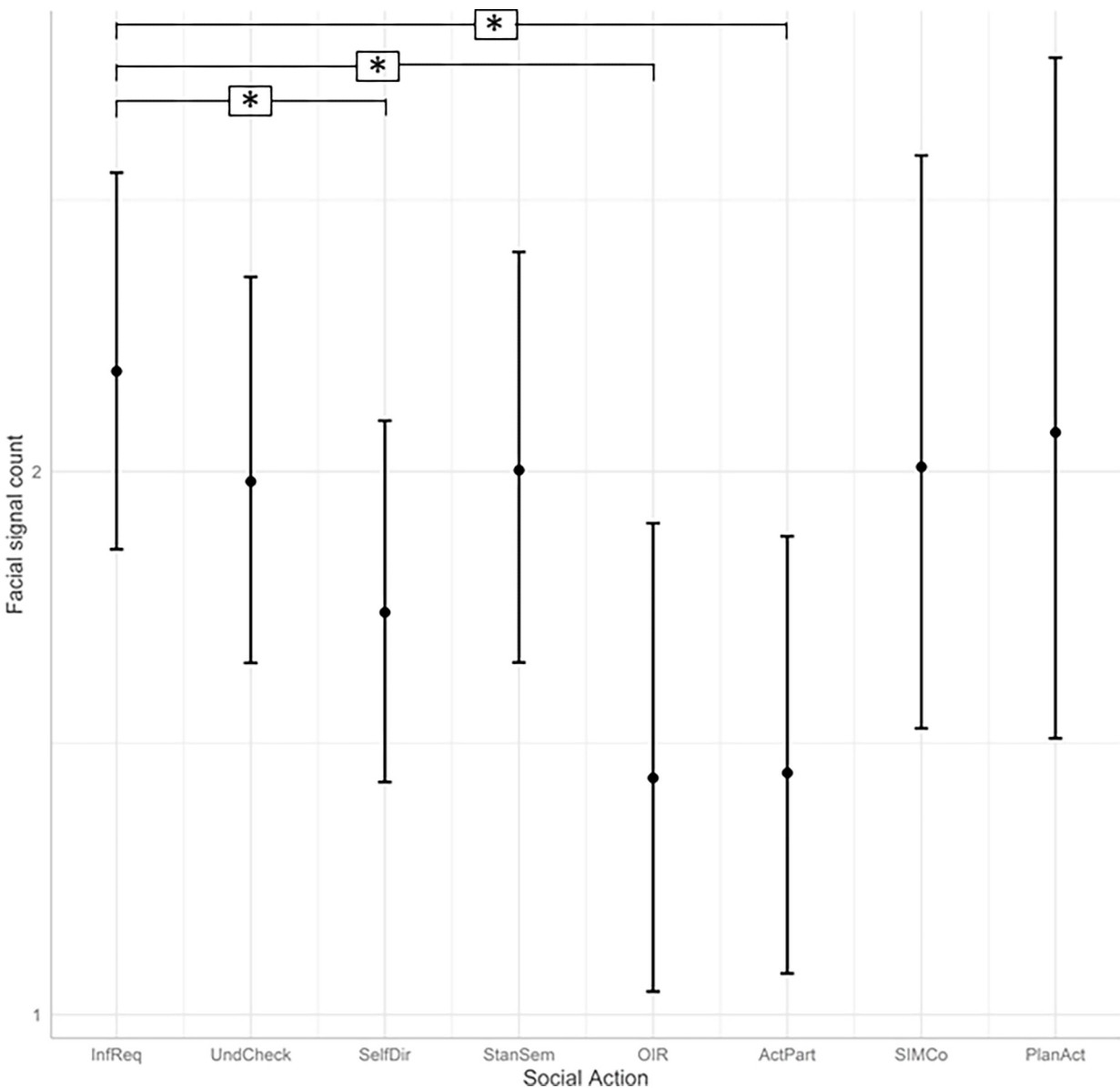

**Fig 2. Predicted facial signal count per social action while holding model terms like utterance count constant.** Social action categories are given on the x-axis, and facial signal counts are indicated on the y-axis. InfReq = Information Requests, UndCheck = Understanding Checks, SelfDir = Self-Directed questions, StanSem = Stance or Sentiment questions, OIR = Other-Initiated Repairs, ActPart = Active Participation questions, SIMCo = questions intended for Structuring, Initiating or Maintaining Conversation, PlanAct = Plans and Actions questions. The model equation was Facial signal count ~ Social action category + Utterance count + (1 | Signal) + (1 | Item).

### Timings of facial signals within social actions

To determine how facial signal onsets primarily distribute across the verbal utterances, and whether there were differences across social actions, we looked at the onset of facial signals relative to the utterance duration (standardised from 0 to 1, with a window of analysis from -1 to 1 to enable visualization facial signal onset distribution before the start of the verbal utterance). Overall, most facial signal onsets occurred around the onset of the verbal utterance. Gaze shifts away from the speaker occurred most before the onset of the utterance, eyebrow frowns prior to or at the beginning of the utterance, whereas eyebrow raises, unilateral eyebrow raises, and

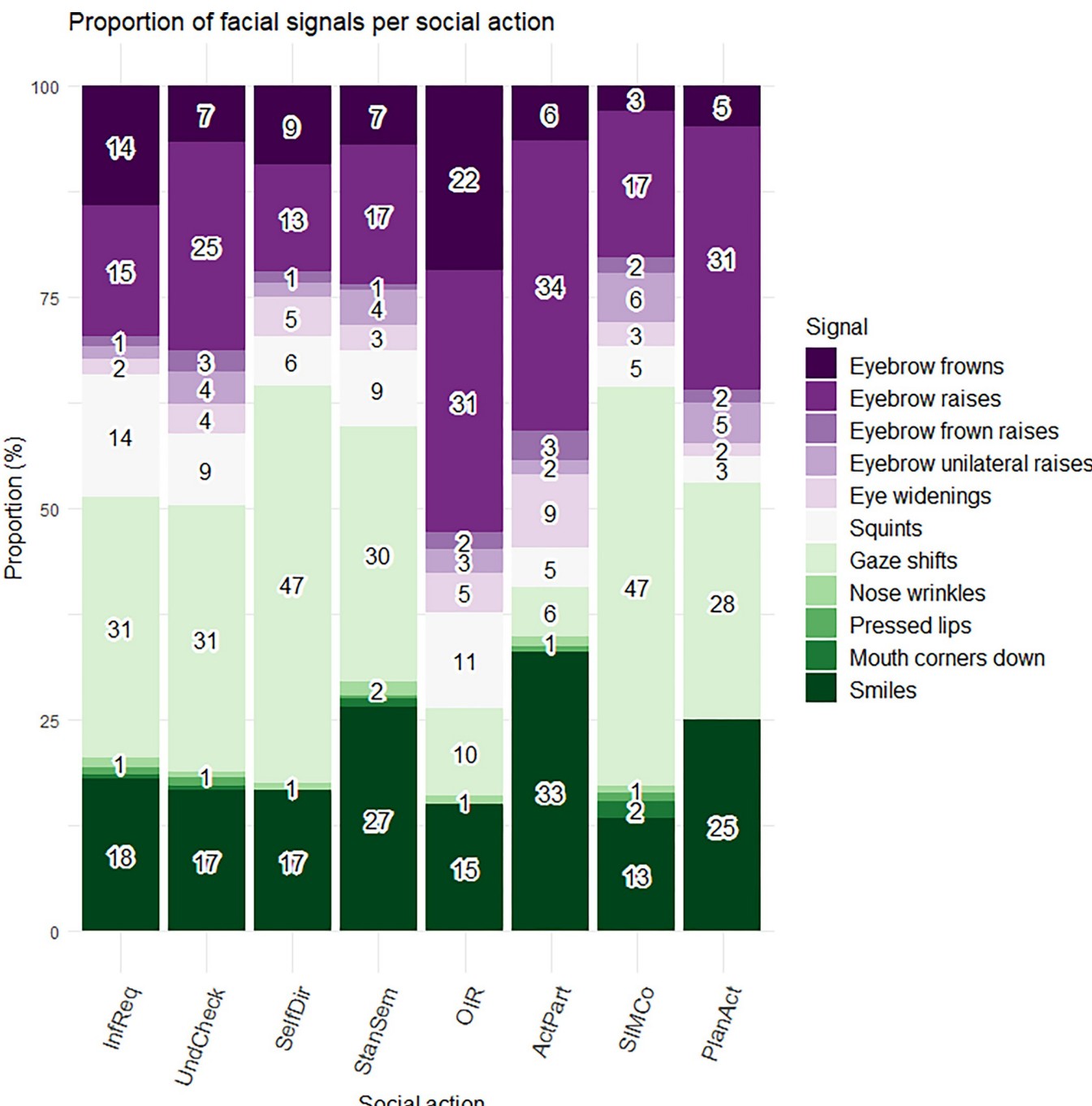

**Fig 3. Proportion of facial signals per social action.** On the x-axis, the proportion is given for each facial signal that occurred together with each social action category out of the respective social action's total number of facial signal occurrences. On the y-axis, we see facial signals split by social action category. InfReq = Information Requests, UndCheck = Understanding Checks, SelfDir = Self-Directed questions, StanSem = Stance or Sentiment questions, OIR = Other-Initiated Repairs, ActPart = Active Participation questions, SIMCo = questions intended for Structuring, Initiating or Maintaining Conversation, PlanAct = Plans and Actions questions.

blinks, often occurred a little after the onset of the utterance. Pressed lips and mouth corners down occurred most near the end of the utterance. Smiles were mostly distributed over the whole utterance (Fig 5).

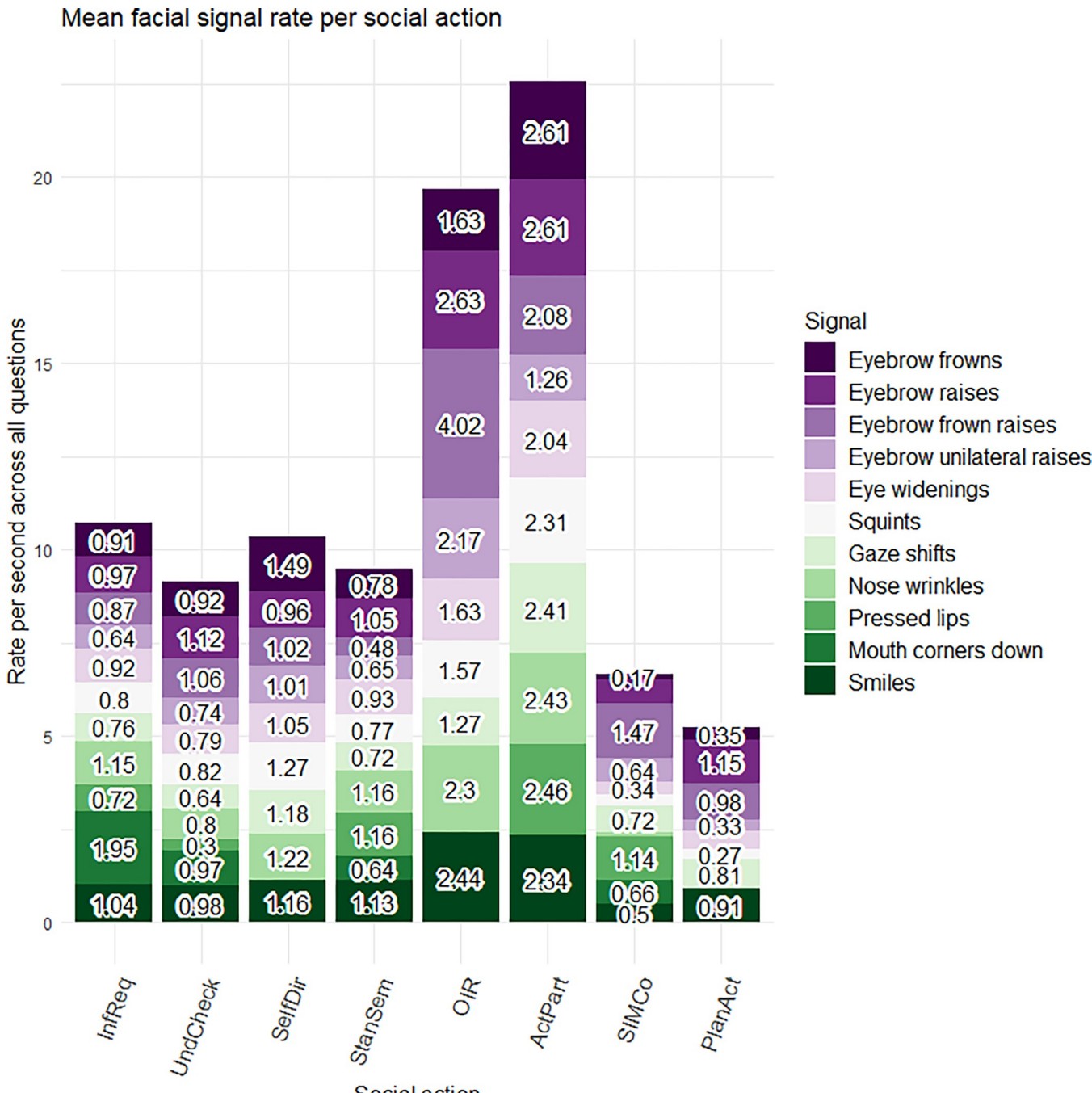

**Fig 4. Mean rate of facial signals per social action.** On the x-axis, the rate per second is given for each facial signal. On the y-axis, we see facial signals split by social action category. InfReq = Information Requests, UndCheck = Understanding Checks, SelfDir = Self-Directed questions, StanSem = Stance or Sentiment questions, OIR = Other-Initiated Repairs, ActPart = Active Participation questions, SIMCo = questions intended for Structuring, Initiating or Maintaining Conversation, PlanAct = Plans and Actions questions.

In relation to social action categories, unilateral eyebrow raises generally occurred around the start of the utterance or a little after across social actions, except for in *Other-Initiated Repairs*, where it occurred before the start of the utterance, and *Plans and Actions questions*,

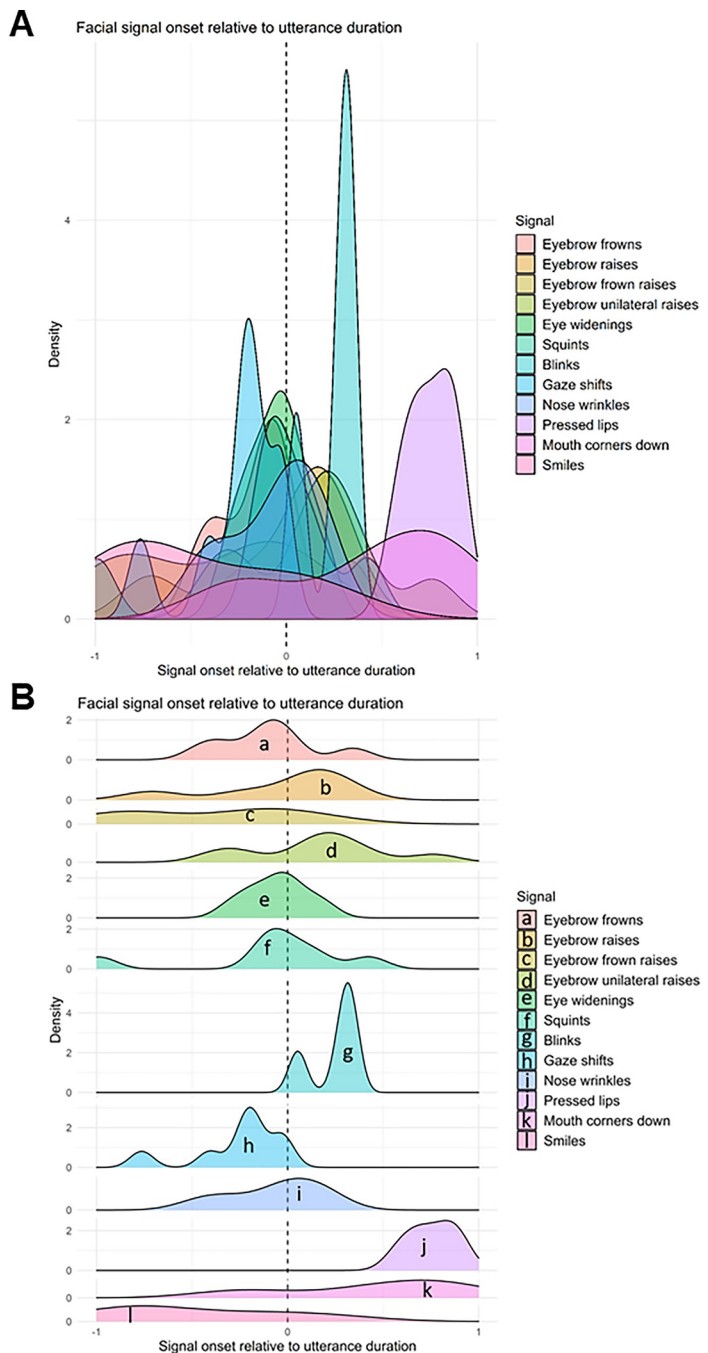

**Fig 5. Overview of facial signal onsets relative to verbal utterance onset.** Panel (A) contains all facial signals plotted on the same y-axis. Panel (B) has a separate y-axis for each specific facial signal. Negative values indicate that the signal onset preceded the start of the verbal utterance, ms = milliseconds.

where it occurred towards the end of the utterance. Nose wrinkles occurred at the start of the utterance in *Information Requests* and *Understanding Checks*, but occurred before the utterance in *Active Participation questions*. No major differences were observed in the timings of the other facial signals (Fig 6).

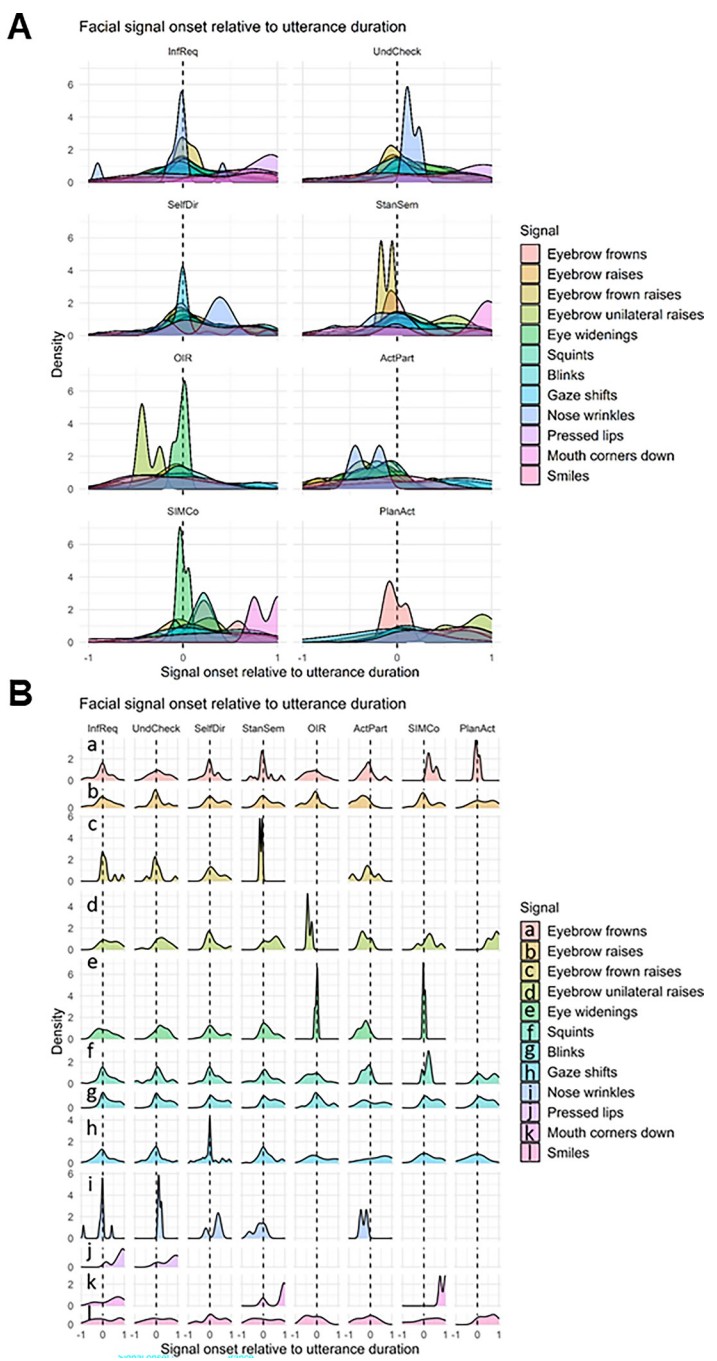

**Fig 6. Overview of facial signal onsets relative to the onset of social actions.** Panel (A) contains all facial signals plotted on the same y-axis. Panel (B) has a separate y-axis for each specific facial signal. Negative values indicate that the signal onset preceded the start of the verbal utterance, ms = milliseconds.

## Temporal organization of facial signals with regard to one another across the different social actions

**Proportion of facial signal sequences.** To capture the sequential patterns of facial signals, we looked at facial signal sequences and selected the most frequent sequences (defined as $n > 4$). This resulted in 12 different frequent facial signal sequences ($n = 164$) with a total of 44 sequences in

*Information Requests*, 25 sequences in *Understanding Checks*, 37 sequences in *Self-Directed questions*, 22 sequences in *Stance or Sentiment questions*, 6 sequences in *Other-Initiated Repairs*, 27 sequences in *Active Participation questions*, 2 sequences in questions intended for *Structuring, Initiating or Maintaining Conversation*, and 1 sequence in *Plans and Actions questions*.

Although there was a small amount of facial signal sequences overall, there were some interesting differences across the social action categories. As shown in Fig 7, where the proportion of frequent facial signals sequences is plotted out of the total amount of sequence instances per social action, *Information Requests* showed a larger proportion of Eyebrow frowns-Squints, *Understanding Checks* showed a larger proportion of Eyebrow raises-Smiles, *Self-Directed questions* showed a larger proportion of Gaze shifts-Smiles, *Stance or Sentiment questions* showed a larger proportion of Eyebrow raises-Eye widenings, and *Active Participation questions* showed a larger proportion of Smiles-Eyebrow raises-Eye widenings.

**Social-action specific clusters of facial signals.** To find out whether the different social actions were distinguishable based on the set of facial signals that accompanied them, we performed a statistical analysis consisting of DT models [70]. These models constructed prediction models from specific groupings of (or single) facial signals to statistically predict whether a verbal utterance was more likely to be one of the eight social action categories.

The analysis was performed on 4134 observations. Results showed six terminal nodes. From the tree, gaze shifts away from the speaker seem to mark both *Information Requests* and *Self-Directed questions*, since both social actions show similar confidence values. In the absence of gaze shifts, smiles most clearly mark *Information Requests*. In the absence of any of the former signals, eyebrow raises mark *Information Requests*, and eye widenings mark *Self-Directed questions*. In the absence of the former signals, eyebrow frowns appear to be very strong markers of *Information Requests*, since they are associated with the highest confidence values for a single social action. Lastly, in the absence of the former signals, eyebrow frown raises seem to mark *Understanding Checks*. Intriguingly, no combinations of facial signals were predicted to mark specific social actions (Fig 8).

The permutation tests (number of simulations = 1000) showed an overall accuracy of 33% on the dataset, which was above chance level (chance level for eight categories = 12.5%).

**Transitional probability between pairs of facial signals over all sequences.** To explore how likely it was that certain facial signals would be adjacent to each other in facial signal sequences (or overlapped) across social actions, we first looked at the count of adjacent facial signal pairs occurring in the same verbal utterance. Results show that eyebrow frowns were often followed by squints, blinks, or gaze shifts. Eyebrow raises were often followed by more raises, eye widenings, blinks, gaze shifts, or smiles. Squints were often followed by eyebrow frowns or blinks. Blinks were often followed by many other facial signals in general, but mostly by more blinks or gaze shifts. Gaze shifts were often followed by eyebrow frowns, raises, squints, blinks, or smiles. Finally, smiles were often followed by eyebrow raises, blinks, or gaze shifts (Fig 9).

Second, we determined the transitional probabilities between each pair of facial signals over all sequences using Markov chains [74, 75]. Smiles and gaze shifts had the most links with other nodes in questions, followed by eyebrow raises and eyebrow frowns. The highest transitional probabilities ($> 0.5$) were observed from pressed lips to eyebrow raises, nose wrinkles to eyebrow frowns, squints to eyebrow frowns, and eyebrow frowns to squints (Fig 10).

## Discussion

This study investigated how conversational facial signals map onto the expression of social actions conveyed through questions. The distribution, timing, and temporal organization of twelve facial signals across eight different social actions was analysed in a rich corpus of naturalistic and dyadic face-to-face Dutch conversations.

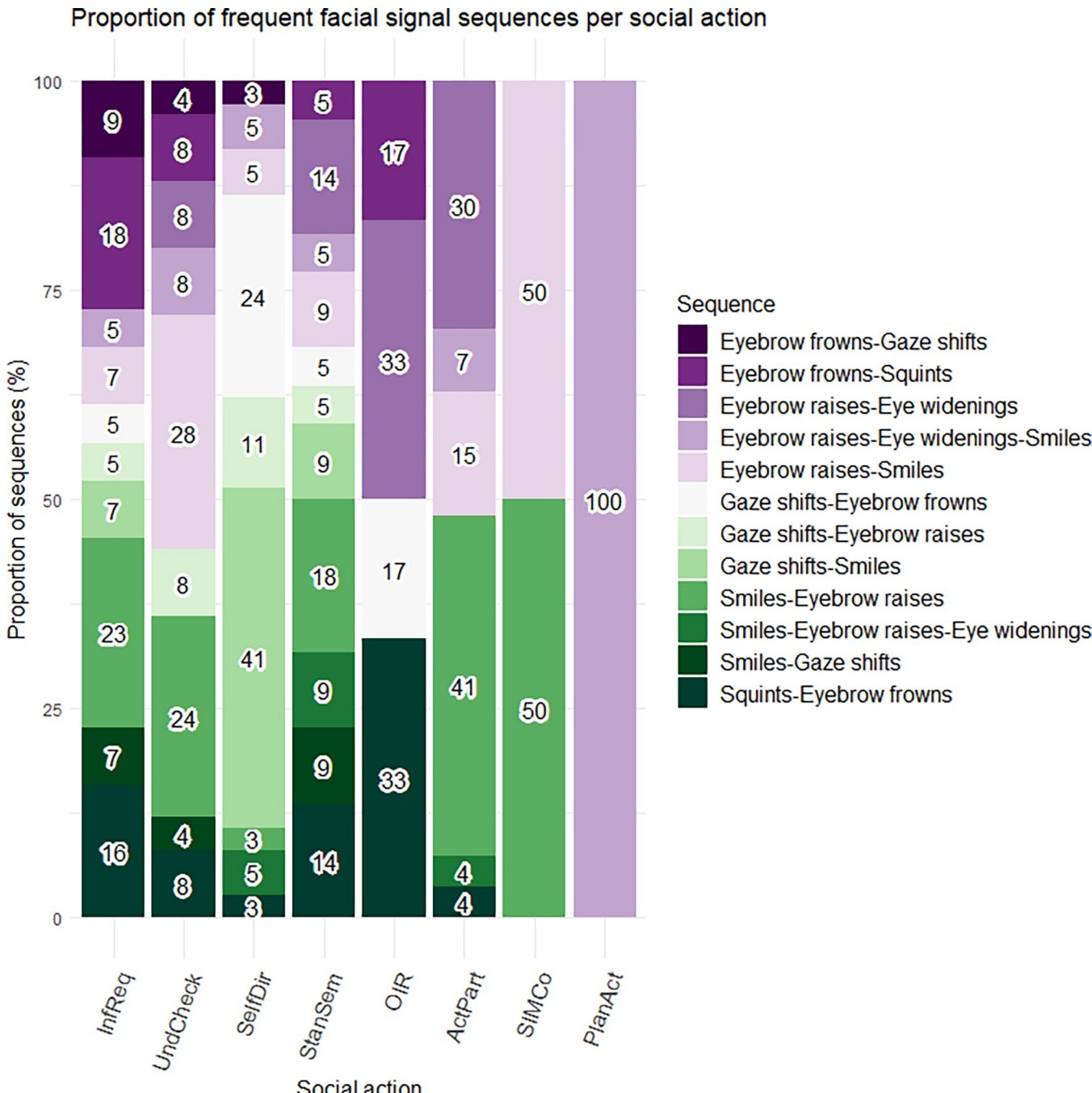

**Fig 7. Proportion of frequent facial signal sequences out of the total amount of sequences observed in each social action.** On the x-axis, we see social action category split by facial signal sequences. On the y-axis, the proportion is given of all facial signal sequences per social action. InfReq = Information Requests, UndCheck = Understanding Checks, SelfDir = Self-Directed questions, StanSem = Stance or Sentiment questions, OIR = Other-Initiated Repairs, ActPart = Active Participation questions, SIMCo = questions intended for Structuring, Initiating or Maintaining Conversation, PlanAct = Plans and Actions questions. Note that questions not accompanied by sequences of visual signals do not form part of the data displayed in this figure.

## Distribution of facial signals across social actions

When looking at the distribution of facial signals across the eight social action categories, most facial signals were found in *Information Requests* ("What is this?"), which may indicate that visual marking is most relevant for requests for information. Furthermore, when looking at

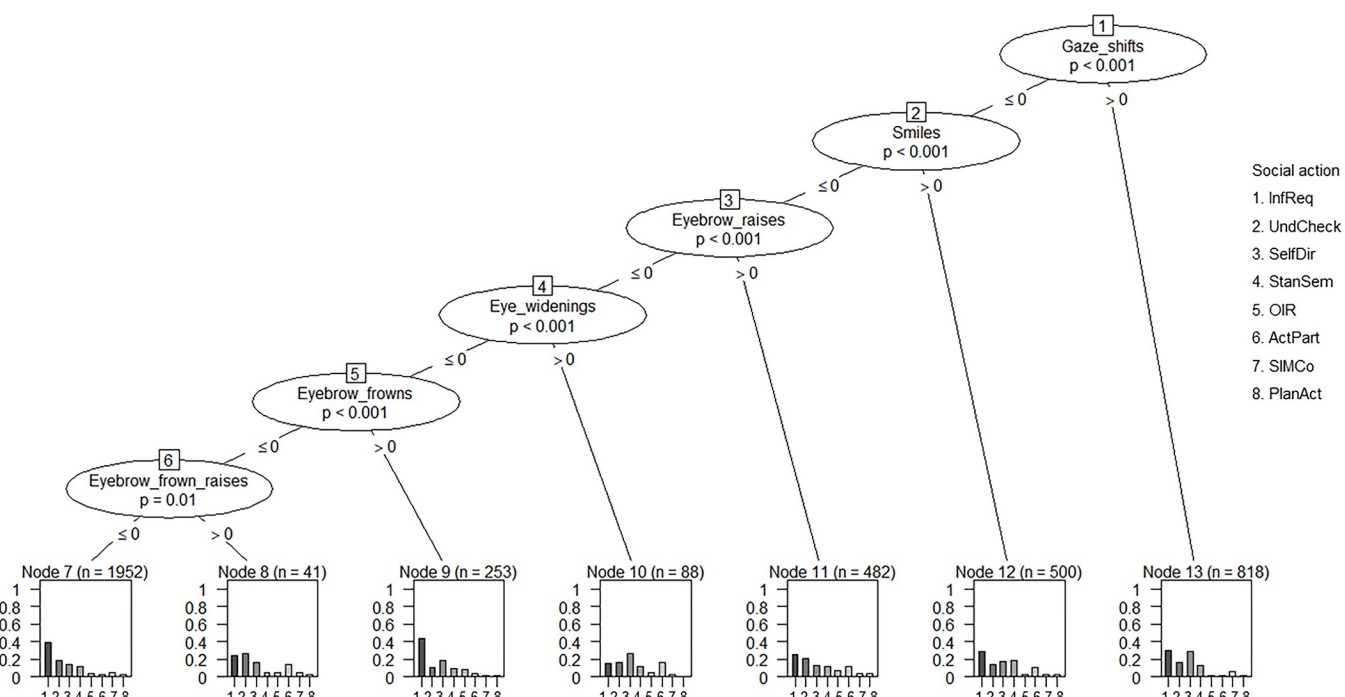

**Fig 8. Conditional inference decision tree.** The decision nodes are represented by circles, and each has a number. They show which facial signals are most strongly associated with the Bonferroni adjusted *p*-value of the dependence test. The input variable to split on is shown by each of these circles, which are divided sequentially (start at the top of the tree). The left and right branches show the cut-off value (i.e., < = 0 means no signal present, > 0 signal present). The bars in the output nodes represent the proportion of social action cases in that node. The bars in order of left to right represent the proportion of: InfReq, UndCheck, SelfDir, StanSem, OIR, ActPart, SIMCo, and PlanAct. Thus, larger bars indicate a higher statistical prediction of an utterance being a specific social action. InfReq = Information Requests, UndCheck = Understanding Checks, SelfDir = Self-Directed questions, StanSem = Stance or Sentiment questions, OIR = Other-Initiated Repairs, ActPart = Active Participation questions, SIMCo = questions intended for Structuring, Initiating or Maintaining Conversation, PlanAct = Plans and Actions questions.

specific facial signals, the data showed that these distribute differently across the social actions. Regarding the proportions of facial signals across social actions, eyebrow frowns and raises often occurred with *Other-Initiated Repairs* ("What?", "Who?"), in agreement with previous research [10, 32, 49, 76]. Furthermore, eyebrow raises often occurred with *Active Participation questions* ("Oh really?"), in agreement with our expectation that eyebrow raises may often serve as back-channels to convey participation [22, 50], or occur in echo questions to help convey news acknowledgments, expressions of surprise, or disbelief [29, 30]. Moreover, gaze shifts away from the addressee often occurred with *Self-Directed questions* ("Now where are my keys?"), in line with the idea that a speaker's gaze aversion may signal still being in the process of something and not requiring active participation of the addressee [39, 42]. Additionally, the finding that smiles often occurred with *Active Participation questions* and *Stance or Sentiment questions* ("Do you think that is fair?") is in line with the idea that smiles may often serve as backchannels to convey participation [22, 50], or may convey irony [47], or genuine positive affect [77].

In terms of comparing the overall frequencies with the duration-standardized analysis, it is important to bear in mind that while certain social actions such as *Information Requests* may be more visually marked overall, the less frequent visual marking that occurs in social actions like *Other-Initiated Repairs* may still be just as important when they do occur. *Other-Initiated Repairs* were typically shorter than *Information Requests* in the corpus, and facial signals could be perceived as more prominent in a shorter utterance.

Our results presented above therefore validate our first hypothesis that social actions would differ with respect to the facial signals they are associated with. The present findings are thus

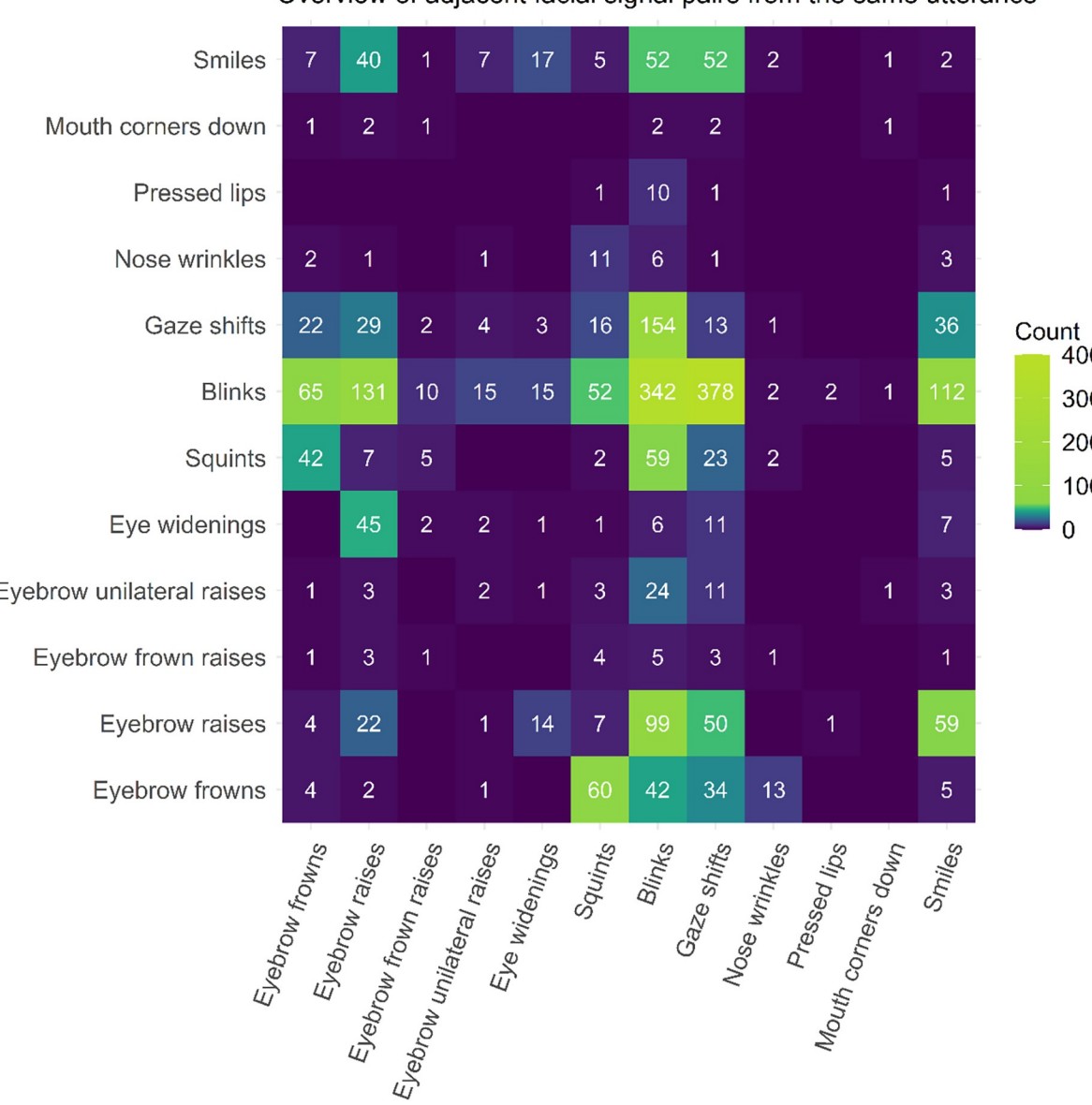

**Fig 9. Overview of facial signal pairs from the same verbal utterance.** The first facial signal is plotted on the x-axis, and the next facial signal on the y-axis. Therefore, the axes show the direction of the transition between facial signal pairs from the same verbal utterance. Count indicates the number of facial signal pairs from the same utterance. When there are no facial signal pairs, the square is left blank.

in line with Nota et al. [33] and build on their analysis that contrasted the conversationally core but broad social action categories 'questions' and 'responses'. The present analysis provided an in-depth, detailed analysis of associations between facial signals and a wide range of different social actions questions themselves can fulfil.

## Timings of facial signals within social actions

When looking at where facial signal onsets primarily distribute in the verbal utterances, most facial signal onsets occurred around the onset of the verbal utterance. This is in line with our second hypothesis, which was motivated by previous findings of Nota et al. [33], and the idea

## Facial signal transition diagram

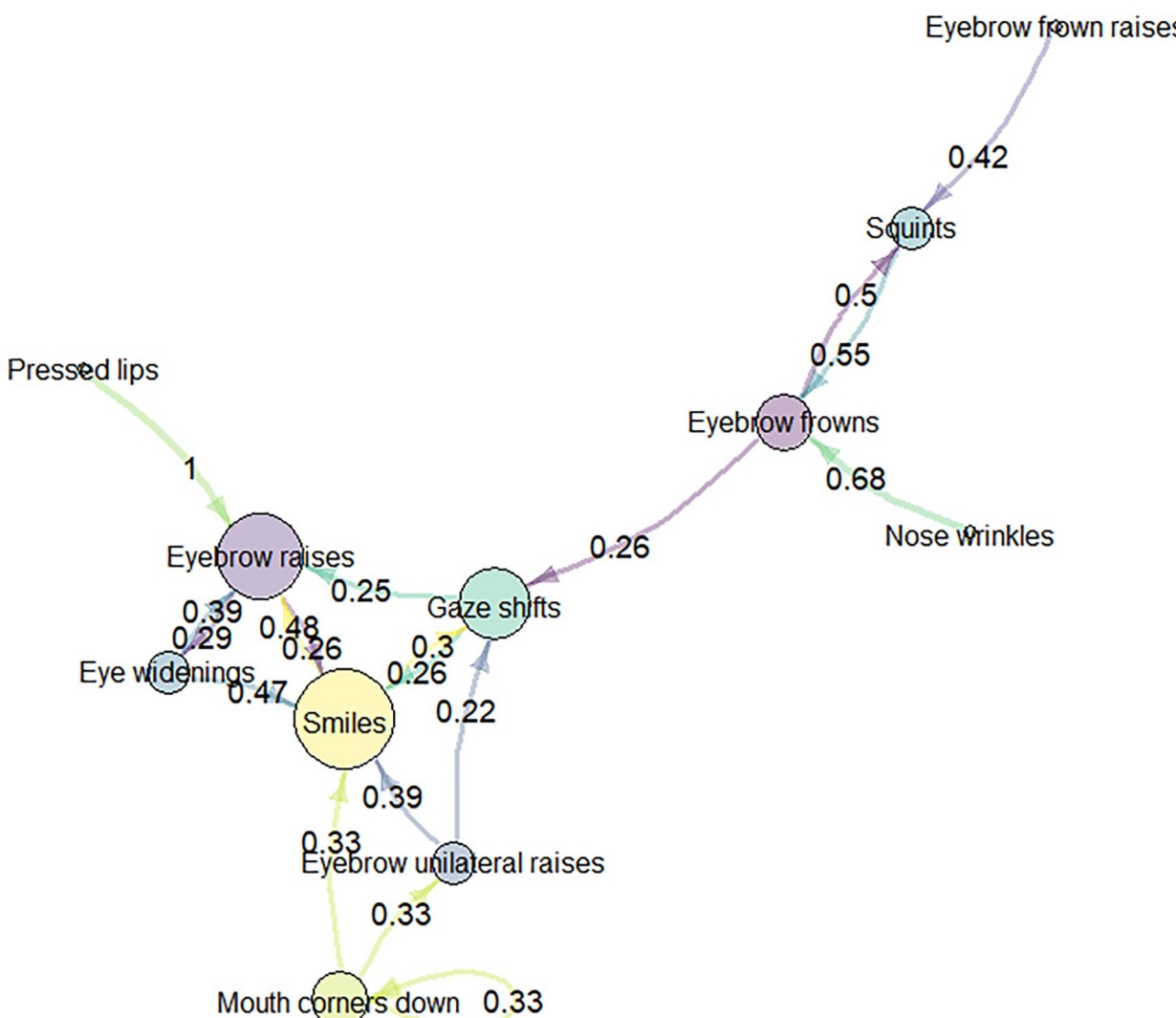

**Fig 10. Facial signal transition probabilities.** Each node represents a facial signal. The node size represents how many different signals may precede or follow (i.e., the more links, the larger the node). Arrow colours are based on their source node colour, thereby showing the direction of the transition between facial signal pairs. Arrows that loop from a facial signal and go back to the same facial signal show a transition between two identical facial signals. Transition probabilities are indicated on the arrows. Transition probabilities below 0.2 were excluded in this diagram.

of early signalling facilitating early action recognition in conversational interaction [3–7]. Like Nota et al. [33], specific mouth movements (pressed lips and mouth corners down) occurred most near the end of the utterance. Diverging from the more typical early signalling may be a signal in itself, such as to indicate irony or sarcasm, since these intentions are typically shown at the end of the utterance for a humoristic effect [46, 47].

Another interesting finding were the observed differences in the timing of facial signal onsets between different social actions. While eyebrow unilateral raises generally occurred

around the start of the utterance (or a little after) across social actions, they occurred before the start of the utterance in *Other-Initiated Repairs*, and occurred towards the end of the utterance in *Plans and Action questions* ("How about lunch together?"). Moreover, nose wrinkles occurred at the start of the utterance in *Information Requests* and *Understanding Checks* ("And you said you wanted to travel next week?"), but occurred before the utterance in *Active Participation questions*. This may indicate that differences in timing of one and the same facial signal may in itself be indicative of social action categories.

## Temporal organization of facial signals with regard to one another across the different social actions

Although only some facial signal sequences were observed in questions, these sequences distributed differently across social actions. Especially interesting was the association of the sequence Eyebrow frowns-Squints with *Information Requests* due to its resemblance to the not-face [40], and the association of Eyebrow raises-Eye widenings with *Stance or Sentiment questions* due to its resemblance to a 'surprise-face' [33, 78]. This is in line with our third hypothesis that known combinations of facial signals would often co-occur, and may indicate that these sequences are most relevant for signalling the aforementioned social actions. Therefore, it may be that while there are only few sequences of facial signals, when there is a specific sequence, it is most likely to be with a particular social action.

When trying to distinguish social actions based on the set of facial signals that accompanied them, eyebrow frowns, together with the absence of gaze shifts, smiles, eyebrow raises and eye widenings, strongly predicted utterances to be *Information Requests*. This shows that eyebrow frowns are strong markers of *Information Requests*, in line with our expected association based on past research [20–22, 24]. Unlike Nota et al. [33], who found that groupings of facial signals could distinguish between question and response social actions using DT models, no combinations of facial signals were found to mark more specific social actions within the broader social action category of questions. Nota et al. [33] examined questions and responses more generally, which meant that the prediction models focused on only two levels to explain associations with the different facial signals instead of eight levels. Thus, it could be that combinations of facial signals play a smaller role when looking at a more detailed level of social action categories.

When exploring whether certain facial signals would occur in a specific adjacency pattern in questions (or overlapped), we observed that smiles and gaze shifts were often adjacent to other signals, followed by eyebrow movements like raises and frowns. Moreover, nose wrinkles were often followed by eyebrow frowns, and eyebrow frowns and squints were often followed by each other. Eyebrow movements therefore seem to be important facial signals for questions. It could be that eyebrow movements are key in signalling different social actions of questions by being in a particular adjacency pattern with other facial signals, but the amount of sequences was too little to perform such an analysis.

## Limitations and future studies

Some methodological limitations were introduced by using artificial cut-offs to overcome the many sub-movements that occurred during (extreme) laughter, and using a video frame rate which made it difficult to code fast consecutive blinks (see also [33]). Social action communication in conversation is incredibly complex and multi-layered, and notoriously difficult to capture in categories. The current approach is thus certainly not without flaws, but it uses a carefully created coding system based on a variety of extant works on social actions in conversation and paying close attention to the social interactional context of utterances when

determining social actions. It is thus the first attempt to systematically quantify social actions in a large body of conversational data, while trying to take account of the complexities and subtleties of human interaction as much as possible.

Participants varied significantly in the content they conveyed, as well as their perspectives, which we interpreted to indicate a high level of comfort with being recorded and therefore to be an accurate reflection of common everyday dialogue. However, different social factors could still have an effect on participant's signalling behaviour, such as the degree of familiarity between speakers, or the fact that the speakers were sitting in a lab while being recorded, potentially causing them to modify their behaviour due to being observed (similar to the 'Hawthorne effect' [79]).

Corpus data inherently involves many intertwined layers of behaviour, which we cannot tease apart without experimental manipulation. Future experimental studies should therefore investigate the exact contribution facial signals make towards quick social action recognition in conversation, to control for other potential factors (e.g., social factors, turn boundaries, interaction with prosody). Furthermore, investigating visual signalling in other group contexts and across non-WEIRD (Western Educated Industrialized Rich and Democratic) societies would be particularly relevant to find out whether the current findings hold in different social and cultural settings [80, 81].

## Conclusion

To conclude, this study demonstrates that facial signals associate with a range of different social actions in conversation, by revealing different distributions and timings of facial signals across social actions, as well as several sequential patterns of facial signals typical for different social actions. Facial signals may thus facilitate social action recognition in multimodal face-to-face interaction.

These findings provide the groundwork for future experimental investigations on the contribution of facial signals on social action recognition. Crucially, our study extends previous work on (individual) facial signals and social actions by involving various social actions from a large dataset of naturalistic, entirely unscripted conversations, while taking into account the social interactional embedding of speakers' behaviour, and using state of the art approaches to analyse the richness of dyadic conversation on many different levels.

## Supporting information

**S1 Appendix. Overview set-up from Nota et al.** [33].
(TIFF)

**S2 Appendix. Summary of the relationship between the 34 CoAct dyads and their conversation quality.**
(DOCX)

## Acknowledgments

We thank Anne-Fleur van Drunen, Guido Rennhack, Hanne van Uden, Josje de Valk, Leah van Oorschot, Maarten van den Heuvel, Mareike Geiger, Marlijn ter Bekke, Pim Klaassen, Rob Evertse, Veerle Kruitbosch and Wieke Harmsen for contributing to the collection and annotation of the corpus data. In addition, we thank Han Sloetjes for technical assistance with ELAN, Jeroen Geerts for support with laboratory facilities, Lucas van Duin and Teun van Gils for providing advice regarding the analyses.

## Author Contributions

**Conceptualization:** Naomi Nota, James P. Trujillo, Judith Holler.

**Data curation:** Naomi Nota.

**Formal analysis:** Naomi Nota.

**Investigation:** Naomi Nota.

**Methodology:** Naomi Nota, James P. Trujillo, Judith Holler.

**Project administration:** Naomi Nota.

**Resources:** Judith Holler.

**Supervision:** James P. Trujillo, Judith Holler.

**Visualization:** Naomi Nota.

**Writing – original draft:** Naomi Nota.

**Writing – review & editing:** Naomi Nota, James P. Trujillo, Judith Holler.

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
