## [Decision Letter · Decision Letter 0]

15 Mar 2023

PONE-D-22-28214Specific facial signals associate with categories of social actions conveyed through questionsPLOS ONE

Dear Dr. Nota,

Thank you for submitting your manuscript to PLOS ONE. After careful consideration, we feel that it has merit but does not fully meet PLOS ONE’s publication criteria as it currently stands. Therefore, we invite you to submit a revised version of the manuscript that addresses the points raised during the review process.

We look forward to receiving your revised manuscript.

Kind regards,

Celia Andreu-Sánchez

Academic Editor

PLOS ONE

Journal Requirements:

This work was supported by an ERC Consolidator grant (#773079, awarded to Judith Holler).

However, funding information should not appear in the Acknowledgments section or other areas of your manuscript. We will only publish funding information present in the Funding Statement section of the online submission form. 

This work was supported by an ERC Consolidator grant https://erc.europa.eu (#773079, awarded to JH). The funders had no role in study design, data collection and analysis, decision to publish, or preparation of the manuscript.

5. We note that Figure 1 includes an image of a participant in the study. 

Reviewers' comments:

Reviewer's Responses to Questions

**Comments to the Author**

1. Is the manuscript technically sound, and do the data support the conclusions?

Reviewer #1: Yes

Reviewer #2: Yes

2. Has the statistical analysis been performed appropriately and rigorously? 

Reviewer #1: Yes

Reviewer #2: N/A

3. Have the authors made all data underlying the findings in their manuscript fully available?

Reviewer #1: Yes

Reviewer #2: Yes

4. Is the manuscript presented in an intelligible fashion and written in standard English?

Reviewer #1: Yes

Reviewer #2: Yes

5. Review Comments to the Author

Reviewer #1: Reviewer’s Comment

This research investigates the association between different facial signals and social actions expressed through questions. The research questions that were asked were:

1) RQ1: Do different social actions result in a different distribution of facial signals?

2) RQ2: How does the timing of facial actions coordinate with verbal utterances in performing social actions?

3) RQ3: In what way do the facial signals of different social actions differ in terms of their temporal organization?

Specific hypotheses that were asked were:

H1: Social actions will differ in accordance with the facial signals that accompany them.

H2: Facial signals are most likely to occur at the beginning of a conversation.

Based on the results presented, the following were the observations presented for different RQs:

1) RQ1 response: The facial signals distribute differently across social signals.

2) RQ2 response: The onset of the facial signal was most common at the moment when the verbal utterance began.

3) RQ3 response: The temporal organization of the facial signal sequences was distributed differently across social actions.

—------------------------------------------------------------------------------------------------------------------

Overall, the paper is well-written and presents a clear and concise argument related to different facial signals associated with different social actions. The methodology is appropriate, the results are statistically significant. The paper makes an important contribution to the existing body of knowledge on the topic. I can also see a wider applicability of the research when having one-on-one conversations and preparing oneself beforehand to come up with appropriate responses.

There are still some comments that need to be addressed for the paper to be published. Here are my comments:

1) In light of the fact that this experiment involved the capture of the faces of participants using cameras and informed consent was obtained from the participants informing them that their facial signals or facial profiles would be captured, how did the author account for the Hawthorne effect? Did the authors consider the impact of the experiment on the accuracy of the data collected?

2) Presently, the study has a skewed distribution of gender, i.e., 51 females and only 17 males. Additionally, the standard deviation is high, i.e., 08 years.

a) Has this high SD affected the timing and distribution of facial signals across social actions?

b) How did the authors account for the gender-based differences in facial signals between males and females, given that literature has found gender differences in emotional expression through the face [1,2]? It is understandable that the component measured here was facial signals and not facial expressions; however, facial signals also include muscle movements that may differ between genders. I believe facial signals will vary across genders.

[1] http://ccat.sas.upenn.edu/plc/communication/soojin.htm

[2] https://www.frontiersin.org/articles/10.3389/fpsyg.2015.01372/full

3) Furthermore, why did the authors not examine dyadic conversions between?

a) Male-male diads

b) Male-female diads

c) Female-female diads

This question is intended to assess the generalizability of the study to a broader population. It would have been interesting to see the timing and distribution of the facial signals across social categories between these genders.

4) In your lines 218 to 220 - “Following the same procedure as for questions transcriptions, interrater reliability for the social action categories was calculated for 10% of the total number of question annotations (n = 686)”.

Considering that the total number of annotated questions was 2082 (lines 205-206), 10% of that number should represent something else? I believe that this needs to be corrected. Or correct me if I am wrong.

5) In line (353) you mentioned that you used holdout cross-validation. How were your training and validation datasets distributed? You should explicitly mention the ratio of the training and validation dataset somewhere in your paper.

6) Please include verbatims of the verbal utterances for different social actions in sections 4.1, 4.2, and 4.3. By doing so, the audience will have a better understanding of the paper. As an example, what verbatim responses occurred during other-initiated repairs during which eyebrows were raised and frowned?

7) Please indicate explicitly in the discussion section how your study validated the two hypotheses you discussed in section 1.1. In doing so, you will make your audience feel connected to the introduction section.

—------------------------------------------------------------------------------------------------------------------

Reviewer #2: The manuscript approaches the topic of facial behavior as a tool for social communication from a perspective that focuses on its pragmatic function. Studies grounded on this perspective and data allowing to test if particular facial behaviors can be reliable and discriminative signals of specific social actions are worthy contributions to the field. Probably because of the technical difficulties involved, studies measuring actual facial behavior are scarce in comparison to studies focused on how facial signals are interpreted. In addition, a pending task in the field has to do with measuring and exploring the temporal dynamics of facial behavior. The study presented in the manuscript does so following a clear and solid rationale, which further improves its relevance. However, I noticed a few issues that, in my opinion, worth to be addressed:

- How were the participants of the study recruited? Were any specific criteria followed? I may have missed it, but I have not seen this addressed in the method section. In Trujillo and Holler (2021) it is reported that information about the relationship between the individuals within each dyad was collected. This makes sense, since the previous relationship could have an impact in the depth and nature of their shared knowledge and, therefore, in what facial signals and utterances could appear and with what frequency during interactions (e.g. I would expect irony to be more likely in a context in which a trust relationship is already built). Therefore, I think this information worth to be used as a control variable in the analyses or, at least, for providing a descriptive summary that would help the reader to assess features of the sample that could have a relevant impact in the behaviors observed.

- When presenting the coding procedure of facial signals (section 2.1.1.3), the Pearson correlation or the standard error of measurement are mentioned as indexes used to quantify the reliability of the coding of timing. However, the magnitudes obtained for them are not provided. I wonder what the purpose of mentioning them is if there is no intention to provide the corresponding results.

- The generalized linear mixed effect models fitted (section 2.2.1.1.) included random intercepts. Why are the predicting variables (social action category and utterance count per social action) included as fixed effects? If I am not mistaken, since each participant can show multiple social actions, he/she can contribute with several data points. This adds interdependencies in the data that could be properly taken into account including random, and not fixed slopes for the predicting variables.

Minor issues

There are two entries in the references list numbered as 33. I am not sure if this is intended, since there are several other entries in the list that share the assigned number. If it is not intended, I suggest checking the list and, if corrections are introduced as a result, the corresponding cites in the text should be revised too (e.g. Nota et al. cite should end with [34] instead of [33]).

6. PLOS authors have the option to publish the peer review history of their article (what does this mean?). If published, this will include your full peer review and any attached files.

Reviewer #1: **Yes: **Mritunjay Kumar

Reviewer #2: No

---

## [Author Response · Author response to Decision Letter 0]

25 May 2023

Dear Celia Andreu-Sánchez,

We hereby submit the revised version of our manuscript, titled “Specific facial signals associate with categories of social actions conveyed through questions”, for your consideration.

We first would like to express our gratitude to you and the reviewers for providing valuable feedback and guidance that helped us enhance the manuscript. We have made significant revisions to the manuscript to address the concerns that were expressed by you and the reviewers.

First, we have ensured that the manuscript meets PLOS ONE’s style requirements and have removed the funder information from the main text. We have uploaded our study’s minimal dataset on the Open Science Framework project website https://osf.io/u59kb/?view_only=d2b7f98f7ba646d69c8afd5cf09e4b2e, and also refer to this in our manuscript in the Data availability statement. We have changed Figure 1 by masking any identifying information of the individuals, since consent forms were retrieved at the time of collecting the corpus, and cannot be asked for PLOS ONE specifically anymore. We have adjusted the figure legend respectively, and included captions for our Appendix materials.

Second, we have changed the colours of Figure 3, 4, and 7, to increase readability of our figures. While changing these figure colours, we corrected a mistake in our analysis script by adding one missing grouping variable for calculating the mean rate of facial signals per social action. We have changed the corresponding text and figure for this specific analysis (Figure 4, mean rate of facial signals per social action), as well as updated our analysis script on the Open Science Framework Project website.

Third, we addressed the concern that social factors may have affected our results (such as familiarity or similarities between speakers or other individual differences across speakers) by clarifying our design and discussing this point in more detail. We additionally provided a descriptive summary of the relationship and experienced conversation quality between speakers in the Appendix, and added the complete questionnaire results on the Open Science Framework project website. We have also clarified other methodology details throughout our manuscript. 

We are convinced that these additions and revisions further strengthened our submission.

Detailed answers to the points suggested by the reviewers are added below, together with the original comments. We have marked our responses in red, with the additions to the manuscript indicated in quotes. The updated manuscript shows our revisions marked with Track Changes.

We look forward to hearing your assessment of our revised manuscript. 

Naomi Nota, James Trujillo, Judith Holler

 

Reviewer #1: Reviewer’s Comment

This research investigates the association between different facial signals and social actions expressed through questions. The research questions that were asked were:

1) RQ1: Do different social actions result in a different distribution of facial signals?

2) RQ2: How does the timing of facial actions coordinate with verbal utterances in performing social actions?

3) RQ3: In what way do the facial signals of different social actions differ in terms of their temporal organization?

Specific hypotheses that were asked were:

H1: Social actions will differ in accordance with the facial signals that accompany them.

H2: Facial signals are most likely to occur at the beginning of a conversation.

Based on the results presented, the following were the observations presented for different RQs:

1) RQ1 response: The facial signals distribute differently across social signals.

2) RQ2 response: The onset of the facial signal was most common at the moment when the verbal utterance began.

3) RQ3 response: The temporal organization of the facial signal sequences was distributed differently across social actions.

—------------------------------------------------------------------------------------------------------------------

Overall, the paper is well-written and presents a clear and concise argument related to different facial signals associated with different social actions. The methodology is appropriate, the results are statistically significant. The paper makes an important contribution to the existing body of knowledge on the topic. I can also see a wider applicability of the research when having one-on-one conversations and preparing oneself beforehand to come up with appropriate responses.

There are still some comments that need to be addressed for the paper to be published. Here are my comments:

1) In light of the fact that this experiment involved the capture of the faces of participants using cameras and informed consent was obtained from the participants informing them that their facial signals or facial profiles would be captured, how did the author account for the Hawthorne effect? Did the authors consider the impact of the experiment on the accuracy of the data collected?

We thank the reviewer for raising this point. We believe our results are as close as possible to natural conversation when recording it (in any setting), since we asked participants to interact freely, resulting in entirely unscripted and unprompted conversation. The participant dyads often discussed topics that were very personal or even private, leading us to believe that they felt comfortable enough to behave naturally. 

To address this point, we have added a sentence to our methods section: “We are confident that the data collected in this corpus reflects common everyday dialogue as participants varied in the content they conveyed and expressed diverse perspectives on comparable themes while being recorded, indicating a high level of comfort.” (p.8-9, lines 174-177).

Importantly, the information participants received prior to being recording did not mention facial behaviour or any other form of visual behaviour. Instead, the emphasis was on aiming to capture casual conversation, and asked them to ignore the recording devices the best they can. Of course, we cannot rule out that the presence of the cameras may have influenced participants’ visual behaviour to some extent, but since the behaviour looked natural, reproduced certain findings that have been reported in the literature before (e.g. on the correlations between eyebrow movements and questions, or specific combinations of facial signals), we are confident that any such influences would not have led to significant distortions of the data. Moreover, there is no alternative, since due to GDPR guidelines it is not allowed to record participants without their prior consent. 

Additionally, we addressed the Hawthorne effect more explicitly in the discussion: “Participants varied significantly in the content they conveyed, as well as their perspectives, which we interpreted to indicate a high level of comfort with being recorded and therefore to be an accurate reflection of common everyday dialogue. However, different social factors could still have an effect on participant’s signalling behaviour, such as the degree of familiarity between speakers, or the fact that the speakers were sitting in a lab while being recorded, potentially causing them to modify their behaviour due to being observed (similar to the ‘Hawthorne effect’ [79]). Corpus data inherently involves many intertwined layers of behaviour, which we cannot tease apart without experimental manipulation. Future experimental studies should therefore investigate the exact contribution facial signals make towards quick social action recognition in conversation, to control for other potential factors (e.g., social factors, turn boundaries, interaction with prosody). Furthermore, investigating visual signalling in other group contexts and across non-WEIRD (Western Educated Industrialized Rich and Democratic) societies would be particularly relevant to find out whether the current findings hold in different social and cultural settings [80,81].” (p. 33-34, lines 689-701). 

2) Presently, the study has a skewed distribution of gender, i.e., 51 females and only 17 males. Additionally, the standard deviation is high, i.e., 08 years.

a) Has this high SD affected the timing and distribution of facial signals across social actions?

b) How did the authors account for the gender-based differences in facial signals between males and females, given that literature has found gender differences in emotional expression through the face [2,3]? It is understandable that the component measured here was facial signals and not facial expressions; however, facial signals also include muscle movements that may differ between genders. I believe facial signals will vary across genders.

[2] http://ccat.sas.upenn.edu/plc/communication/soojin.htm

[3] https://www.frontiersin.org/articles/10.3389/fpsyg.2015.01372/full

We agree that people vary a lot in visual signal production and perception, which may indeed translate to differences in conversational facial signalling. However, we feel that emphasizing gender differences would potentially reinforce gender stereotypes [1], which we would very much prefer to avoid in light of the current gender debates, which conceives of gender as a dimension (or in fact several) rather than a categorical construct. Additionally, at the time that the corpus data was being collected, gender was still defined as a binary construct, which may not accurately reflect the way all of the participants identify. Therefore, we do not feel that our data would allow a meaningful in-depth analysis into gender differences.

To address the fact that more intensive research is needed to drill deeper into the patterns we have uncovered here by looking at different social factors (such as familiarity or similarities between speakers or other individual differences across speakers), we have added a sentence to the discussion, for which we refer the reviewer to our response to point 1. However, an in-depth analysis that does justice to these different social factors would go beyond the current study.

 [1] https://doi.org/10.1146/annurev-psych-122216-011719

3) Furthermore, why did the authors not examine dyadic conversions between?

a) Male-male diads

b) Male-female diads

c) Female-female diads

This question is intended to assess the generalizability of the study to a broader population. It would have been interesting to see the timing and distribution of the facial signals across social categories between these genders.

We thank the reviewer for these suggestions. It may be that likeness to one conversational partner may change the interaction, but we believe that this may be related to more levels than gender similarity. Therefore, we do not feel that simplifying the many social factors that are involved during the interactions by focusing on gender alone would be sufficient to disentangle the many intertwined behaviours that may play a role in conversational interactions. We refer to our previous responses to point 1 and 2, where we further address this concern. Finally, while we agree that additional social factors would be interesting to investigate, we also believe that such an extensive analysis is beyond the scope of the current manuscript.

4) In your lines 218 to 220 - “Following the same procedure as for questions transcriptions, interrater reliability for the social action categories was calculated for 10% of the total number of question annotations (n = 686)”.

Considering that the total number of annotated questions was 2082 (lines 205-206), 10% of that number should represent something else? I believe that this needs to be corrected. Or correct me if I am wrong.

We have adjusted the sentence to correctly represent the number of annotations and to clarify the methodology (p. 11, lines 225-230).

“A subset of 2082 questions were coded for their social action category. The detailed coding scheme for the social action categories was developed for a larger project that the present study is part of, and was inspired by a combination of previous categorizations [10,12,58,65,66]. We took into account the sequential position and form of the social actions in conversation, state of the common ground between speakers, communicative intention, as well as the result of the speaker’s utterance on the addressee. This resulted in eight discrete social action categories”

5) In line (353) you mentioned that you used holdout cross-validation. How were your training and validation datasets distributed? You should explicitly mention the ratio of the training and validation dataset somewhere in your paper.

We used the default value for distributing the training and validation datasets. Therefore, the fraction of data to keep for training was 0.8. We have added the following sentence: “We used the default value for distributing the training and validation datasets, resulting in a data fraction of 0.8 for the training dataset, and 0.2 for the validation dataset.” (p. 20, lines 392-394).

6) Please include verbatims of the verbal utterances for different social actions in sections 4.1, 4.2, and 4.3. By doing so, the audience will have a better understanding of the paper. As an example, what verbatim responses occurred during other-initiated repairs during which eyebrows were raised and frowned?

We have repeated the examples of the social action categories in the discussion to make interpretation of the results more comprehensible.

7) Please indicate explicitly in the discussion section how your study validated the two hypotheses you discussed in section 1.1. In doing so, you will make your audience feel connected to the introduction section.

We have added a more explicit description of these validations by adding the following sentences: “Our results presented above therefore validate our first hypothesis that social actions would differ with respect to the facial signals they are associated with.” (p. 30, lines 620-621),

“When looking at where facial signal onsets primarily distribute in the verbal utterances, most facial signal onsets occurred around the onset of the verbal utterance. This is in line with our second hypothesis, which was motivated by previous findings of Nota et al. [33], and the idea of early signalling facilitating early action recognition in conversational interaction [3–7]. (p.30-31, lines 630-632), 

“This is in line with our third hypothesis that known combinations of facial signals would often co-occur, and may indicate that these sequences are most relevant for signalling the aforementioned social actions.” (p. 31-32, lines 654-656).

—------------------------------------------------------------------------------------------------------------------

Reviewer #2: The manuscript approaches the topic of facial behavior as a tool for social communication from a perspective that focuses on its pragmatic function. Studies grounded on this perspective and data allowing to test if particular facial behaviors can be reliable and discriminative signals of specific social actions are worthy contributions to the field. Probably because of the technical difficulties involved, studies measuring actual facial behavior are scarce in comparison to studies focused on how facial signals are interpreted. In addition, a pending task in the field has to do with measuring and exploring the temporal dynamics of facial behavior. The study presented in the manuscript does so following a clear and solid rationale, which further improves its relevance. However, I noticed a few issues that, in my opinion, worth to be addressed:

- How were the participants of the study recruited? Were any specific criteria followed? I may have missed it, but I have not seen this addressed in the method section. In Trujillo and Holler (2021) it is reported that information about the relationship between the individuals within each dyad was collected. This makes sense, since the previous relationship could have an impact in the depth and nature of their shared knowledge and, therefore, in what facial signals and utterances could appear and with what frequency during interactions (e.g. I would expect irony to be more likely in a context in which a trust relationship is already built). Therefore, I think this information worth to be used as a control variable in the analyses or, at least, for providing a descriptive summary that would help the reader to assess features of the sample that could have a relevant impact in the behaviors observed.

We thank the reviewer for their useful comments and suggestions. We based our analyses on the same participants as Trujillo and Holler (2021), which we refer to in the first paragraph in the Methods section. 

We have now added a more detailed descriptive summary (p.9-10, lines 194-205): “Informed consent was obtained in written form before and after filming. Before the study, participants were asked to fill in a demographics questionnaire. At the end of the study, information was collected about the relationship between the speakers in the dyads and their conversation quality (see S2 B Appendix for a summary of the relationship between speakers and their conversation quality, and the complete questionnaire results on the Open Science Framework project website https://osf.io/u59kb/?view_only=d2b7f98f7ba646d69c8afd5cf09e4b2e), as well as the Dutch version of the Empathy Quotient [53,54], the Fear of Negative Evaluation scale [55], and an assessment of explicit awareness of the experimental aim. Information from these questionnaires were collected for a wider range of studies, but are not discussed in the current study, since they were not deemed relevant for the analysis at hand. Participants were compensated with 18 euros. The corpus study was approved by the Ethics Committee of the Social Sciences department of the Radboud University Nijmegen (ethic approval code ECSW 2018-124).”

Additionally, we provided a summary of the relationship and experienced conversation quality between speakers in the Appendix, and on the Open Science Framework project website (p.46, lines 900-909): “

S2 Appendix B. Summary of the relationship between the 34 CoAct dyads and their conversation quality.

Min length relationship Mean closenessa Mean conversation quality Platonic relationshipb

4 months 5.98 6.15 “yes” = 57

“no” = 11

Note. The scale for closeness and conversation quality ranged from 1 (not close/poor conversation quality) – 7 (very close/excellent conversation quality).

a Because one speaker from a dyad did not completely fill in the questionnaire, there is one missing value for Mean closeness.

b One dyad differed in their response to the question whether they shared more than a platonic relationship.”

- When presenting the coding procedure of facial signals (section 2.1.1.3), the Pearson correlation or the standard error of measurement are mentioned as indexes used to quantify the reliability of the coding of timing. However, the magnitudes obtained for them are not provided. I wonder what the purpose of mentioning them is if there is no intention to provide the corresponding results.

We performed these analyses to assess how precise the annotations were in terms of timing. We have added an overview of these results to the aforementioned section (p.14-15, lines 293-309):

“A similar procedure as for questions and social action transcriptions was used to calculate interrater reliability on approximately 1% of the total data. In addition, we computed convergent reliability for annotation timing by using a Pearson’s correlation (r), standard error of measurement (SeM), and the mean absolute difference (Mabs, in ms) of signal onsets, to assess how precise the annotations were in terms of timing, if there was enough data to compare. All included facial signals from the paired comparisons showed an average raw agreement of 76% and an average kappa of 0.96, indicating almost perfect agreement. When there was enough data to perform a Pearson’s correlation, all signals showed r = 1 with a p < .0001, indicating a strong correlation. There was not enough data to perform a correlation for eyebrow frown raises, nose wrinkles, and mouth corners down. Results are shown in Table 3 (for more details on the facial signals reliability calculations, see Nota et al. [33]).

Table 3. Overview of facial signal reliability scores [33].

Note. agr = raw agreement [62,63], k = Cohen’s kappa [64], SeM = standard error of measurement, Mabs = mean absolute difference (ms).

a One video frame was equivalent to 40 ms. Thus, we considered the variance based on the reliability of the signals shown by SeM and Mabs as very low (and therefore very precise) when < 40, low when < 80, moderate < 160, and high < 160 (least precise).”

- The generalized linear mixed effect models fitted (section 2.2.1.1.) included random intercepts. Why are the predicting variables (social action category and utterance count per social action) included as fixed effects? If I am not mistaken, since each participant can show multiple social actions, he/she can contribute with several data points. This adds interdependencies in the data that could be properly taken into account including random, and not fixed slopes for the predicting variables.

To clarify, the ‘item’ variable included information about the participant number as well as the theme (1,2, or 3) and dyad number. The utterance count was added as a control variable. To address the issue pointed out by the reviewer, and account for multiple social actions that could be produced by each participant and to avoid potential interdependencies, we additionally ran the following models:

Glmer (facial signal count ~ social action + utterance count + (1 + social action | item) + (1 + social action | facial signal)

Glmer (facial signal count ~ social action + utterance count + (1 + social action | item) + (1 | facial signal) 

However, both models resulted in issues of singular fit. Therefore, we did not include random slopes, and kept the minimal model structure that we originally used in our study. 

We addressed these issues by adding the following sentence (p.16, line 330): “We did not add random slopes, nor an interaction between potentially modulating factors, because this resulted in overfitting the model.”

Minor issues

There are two entries in the references list numbered as 33. I am not sure if this is intended, since there are several other entries in the list that share the assigned number. If it is not intended, I suggest checking the list and, if corrections are introduced as a result, the corresponding cites in the text should be revised too (e.g. Nota et al. cite should end with [34] instead of [33]).

> Thank you, we have revised the references accordingly.

---

## [Decision Letter · Decision Letter 1]

20 Jun 2023

Specific facial signals associate with categories of social actions conveyed through questions

PONE-D-22-28214R1

Dear Dr. Nota,

We’re pleased to inform you that your manuscript has been judged scientifically suitable for publication and will be formally accepted for publication once it meets all outstanding technical requirements.

Kind regards,

Celia Andreu-Sánchez

Academic Editor

PLOS ONE

Additional Editor Comments (optional):

Reviewers' comments:

Reviewer's Responses to Questions

**Comments to the Author**

1. If the authors have adequately addressed your comments raised in a previous round of review and you feel that this manuscript is now acceptable for publication, you may indicate that here to bypass the “Comments to the Author” section, enter your conflict of interest statement in the “Confidential to Editor” section, and submit your "Accept" recommendation.

Reviewer #1: All comments have been addressed

Reviewer #2: All comments have been addressed

2. Is the manuscript technically sound, and do the data support the conclusions?

Reviewer #1: Partly

Reviewer #2: Yes

3. Has the statistical analysis been performed appropriately and rigorously? 

Reviewer #1: Yes

Reviewer #2: Yes

4. Have the authors made all data underlying the findings in their manuscript fully available?

Reviewer #1: Yes

Reviewer #2: Yes

5. Is the manuscript presented in an intelligible fashion and written in standard English?

Reviewer #1: Yes

Reviewer #2: Yes

6. Review Comments to the Author

Reviewer #1: Reviewer’s Comment (2)

Previous Comment from the Reviewer (1):

There are still some comments that need to be addressed for the paper to be published.

Here are my comments:

In light of the fact that this experiment involved the capture of the faces of participants using cameras and informed consent was obtained from the participants informing them that their facial signals or facial profiles would be captured, how did the author account for the Hawthorne effect? Did the authors consider the impact of the experiment on the accuracy of the data collected?

Response by the Authors (1)

We thank the reviewer for raising this point. We believe our results are as close as possible to the natural conversation when recording it (in any setting) since we asked participants to interact freely, resulting in entirely unscripted and unprompted conversation. The participant dyads often discussed topics that were very personal or even private, leading us to believe that they felt comfortable enough to behave naturally.

To address this point, we have added a sentence to our methods section: “We are confident that the data collected in this corpus reflects common everyday dialogue as participants varied in the content they conveyed and expressed diverse perspectives on comparable themes while being recorded, indicating a high level of comfort.” (p.8-9, lines 174-177).

Importantly, the information participants received prior to being recording did not mention facial behaviour or any other form of visual behaviour. Instead, the emphasis was on aiming to capture casual conversation, and asked them to ignore the recording devices the best they can. Of course, we cannot rule out that the presence of the cameras may have influenced participants’ visual behaviour to some extent, but since the behaviour looked natural, reproduced certain findings that have been reported in the literature before (e.g. on the correlations between eyebrow movements and questions, or specific combinations of facial signals), we are confident that any such influences would not have led to significant distortions of the data. Moreover, there is no alternative, since due to GDPR guidelines it is not allowed to record participants without their prior consent.

Additionally, we addressed the Hawthorne effect more explicitly in the discussion: “Participants varied significantly in the content they conveyed, as well as their perspectives, which we interpreted to indicate a high level of comfort with being recorded and therefore to be an accurate reflection of common everyday dialogue. However, different social factors could still have an effect on participant’s signalling behaviour, such as the degree of familiarity between speakers, or the fact that the speakers were sitting in a lab while being recorded, potentially causing them to modify their behaviour due to being observed (similar to the ‘Hawthorne effect’ [79]). Corpus data inherently involves many intertwined layers of behaviour, which we cannot tease apart without experimental manipulation. Future experimental studies should therefore investigate the exact contribution facial signals make towards quick social action recognition in conversation, to control for other potential factors (e.g., social factors, turn boundaries, interaction with prosody). Furthermore, investigating visual signalling in other group contexts and across non-WEIRD (Western Educated Industrialized Rich and Democratic) societies would be particularly relevant to find out whether the current findings hold in different social and cultural settings [80,81].” (p. 33-34, lines 689-701).

Response by the reviewer (1)

Dear Authors,

I appreciate your thorough response regarding the Hawthorne effect. The measures you took to ensure naturalness in participant behavior are commendable the inclusion of this topic in your discussion was a positive step, and the future research considerations proposed to highlight a much-needed holistic perspective on this issue. Nevertheless, your study prompts us to consider innovative methodologies that could further minimize such observer effects in future research.

One suggestion could be the use of immersive, naturalistic settings that mimic participants' everyday environments, thus providing a less obtrusive context for data collection. Another possibility could be the extension of recording durations, allowing participants to grow accustomed to the recording process and thus possibly mitigating observer effects over time.

In conclusion, while the authors' approach was well-conceived and carefully executed, the complexities tied to the Hawthorne effect remain a thought-provoking challenge for researchers in this field, necessitating creative methodological advancements for its effective mitigation.

—----------------—--------------------------------------------------------------------------------------------

Previous Comment from the Reviewer (2):

Presently, the study has a skewed distribution of gender, i.e., 51 females and only 17 males. Additionally, the standard deviation is high, i.e., 08 years.

Has this high SD affected the timing and distribution of facial signals across social actions?

How did the authors account for the gender-based differences in facial signals between males and females, given that literature has found gender differences in emotional expression through the face [2,3]? It is understandable that the component measured here was facial signals and not facial expressions; however, facial signals also include muscle movements that may differ between genders. I believe facial signals will vary across genders.

[2] http://ccat.sas.upenn.edu/plc/communication/soojin.htm

[3] https://www.frontiersin.org/articles/10.3389/fpsyg.2015.01372/full

Response by the Authors (2)

We agree that people vary a lot in visual signal production and perception, which may indeed translate to differences in conversational facial signalling. However, we feel that emphasizing gender differences would potentially reinforce gender stereotypes [1], which we would very much prefer to avoid in light of the current gender debates, which conceives of gender as a dimension (or in fact several) rather than a categorical construct. Additionally, at the time that the corpus data was being collected, gender was still defined as a binary construct, which may not accurately reflect the way all of the participants identify. Therefore, we do not feel that our data would allow a meaningful in-depth analysis into gender differences.

To address the fact that more intensive research is needed to drill deeper into the patterns we have uncovered here by looking at different social factors (such as familiarity or similarities between speakers or other individual differences across speakers), we have added a sentence to the discussion, for which we refer the reviewer to our response to point 1. However, an in-depth analysis that does justice to these different social factors would go beyond the current study.

[1] https://doi.org/10.1146/annurev-psych-122216-011719

Response by the reviewer (2)

Dear Authors,

Your response acknowledging the multifaceted nature of gender in today's society is deeply appreciated. The nuanced understanding exhibited in your study adds a layer of depth that demonstrates your comprehensive grasp on the subject matter.

While I understand your desire to avoid reinforcing gender stereotypes, it may be beneficial to your audience to lightly touch on the gender imbalance in your participant group. The inclusion of such a discussion could offer a more holistic view of the potential variables at play in your study.

Your decision to delve into the impact of social factors, such as speaker familiarity and individual differences, adds an intriguing dimension to your research. However, it does bring up a point of methodological interest. If participants were selected due to certain shared traits or relationships, this would align with 'purposive sampling' methods. If that is the case, it might be useful to clarify this in your methodology section for the benefit of your readers.

—---------------------------------------------------------------------------------------------------------------

Previous Comment from the Reviewer (3):

In your lines 218 to 220 - “Following the same procedure as for questions transcriptions, interrater reliability for the social action categories was calculated for 10% of the total number of question annotations (n = 686)”. Considering that the total number of annotated questions was 2082 (lines 205-206), 10% of that number should represent something else? I believe that this needs to be corrected. Or correct me if I am wrong.

Response by the Authors (3)

We have adjusted the sentence to correctly represent the number of annotations and to clarify the methodology (p. 11, lines 225-230). “A subset of 2082 questions were coded for their social action category. The detailed coding scheme for the social action categories was developed for a larger project that the present study is part of, and was inspired by a combination of previous categorizations [10,12,58,65,66]. We took into account the sequential position and form of the social actions in conversation, state of the common ground between speakers, communicative intention, as well as the result of the speaker’s utterance on the addressee. This resulted in eight discrete social action categories”

Response by the reviewer (3)

Thank you.

—-------------------------------------------------------------------------------------------------------------

Previous Comment from the Reviewer (4):

In line (353) you mentioned that you used holdout cross-validation. How were your training and validation datasets distributed? You should explicitly mention the ratio of the training and validation dataset somewhere in your paper.

Response by the Authors (4)

We used the default value for distributing the training and validation datasets. Therefore, the fraction of data to keep for training was 0.8. We have added the following sentence: “We used the default value for distributing the training and validation datasets, resulting in a data fraction of 0.8 for the training dataset, and 0.2 for the validation dataset.” (p. 20, lines 392-394).

Response by the reviewer (4)

Thank you.

—-------------------------------------------------------------------------------------------------------------

Previous Comment from the Reviewer (5):

Please include verbatims of the verbal utterances for different social actions in sections 4.1, 4.2, and 4.3. By doing so, the audience will have a better understanding of the paper. As an example, what verbatim responses occurred during other-initiated repairs during which eyebrows were raised and frowned?

Response by the Authors (5)

We have repeated the examples of the social action categories in the discussion to make interpretation of the results more comprehensible.

Response by the reviewer (5)

Ok

—---------------------------------------------------------------------------------------------------------------

Previous Comment from the Reviewer (6):

Please indicate explicitly in the discussion section how your study validated the two hypotheses you discussed in section 1.1. In doing so, you will make your audience feel connected to the introduction section.

Response by the Authors (6)

We have added a more explicit description of these validations by adding the following sentences: “Our results presented above therefore validate our first hypothesis that social actions would differ with respect to the facial signals they are associated with.” (p. 30, lines 620-621),

Response by the reviewer (6)

Thank You

Reviewer #2: All the issues raised have been properly addressed by the authors. The only point that still puzzles me has to do with interdependence of data points when fitting the generalized linear mixed effect model. Each participant can contribute with a different number of events, which could bias the estimation of the parameters of the model. This problem is partially taken into account by including random intercepts, but not random slopes. However, the authors tried to fit a model with random slopes and they report that it did not yield proper estimates in the manuscript, so the reader would have information enough to be aware of this problem.

7. PLOS authors have the option to publish the peer review history of their article (what does this mean?). If published, this will include your full peer review and any attached files.

Reviewer #1: **Yes: **Mritunjay Kumar

Reviewer #2: No

---

## [Editor Report · Acceptance letter]

11 Jul 2023

PONE-D-22-28214R1 

Specific facial signals associate with categories of social actions conveyed through questions 

Dear Dr. Nota:

I'm pleased to inform you that your manuscript has been deemed suitable for publication in PLOS ONE. Congratulations! Your manuscript is now with our production department. 

Kind regards, 

on behalf of

Dr. Celia Andreu-Sánchez 

Academic Editor

PLOS ONE